# The need for high-quality oocyte mitochondria at extreme ploidy dictates mammalian germline development

**Marco Colnaghi[1,2], Andrew Pomiankowski[1,2]\*, Nick Lane[1,2]**

[1]CoMPLEX, University College London, London, United Kingdom; [2]Department of Genetics, Evolution and Environment, University College London, London, United Kingdom

**Abstract** Selection against deleterious mitochondrial mutations is facilitated by germline processes, lowering the risk of genetic diseases. How selection works is disputed: experimental data are conflicting and previous modeling work has not clarified the issues; here, we develop computational and evolutionary models that compare the outcome of selection at the level of individuals, cells and mitochondria. Using realistic *de novo* mutation rates and germline development parameters from mouse and humans, the evolutionary model predicts the observed prevalence of mitochondrial mutations and diseases in human populations. We show the importance of organelle-level selection, seen in the selective pooling of mitochondria into the Balbiani body, in achieving high-quality mitochondria at extreme ploidy in mature oocytes. Alternative mechanisms debated in the literature, bottlenecks and follicular atresia, are unlikely to account for the clinical data, because neither process effectively eliminates mitochondrial mutations under realistic conditions. Our findings explain the major features of female germline architecture, notably the longstanding paradox of over-proliferation of primordial germ cells followed by massive loss. The near-universality of these processes across animal taxa makes sense in light of the need to maintain mitochondrial quality at extreme ploidy in mature oocytes, in the absence of sex and recombination.

**\*For correspondence:**
a.pomiankowski@ucl.ac.uk

**Competing interests:** The authors declare that no competing interests exist.

## Introduction

In mammals, mitochondrial gene sequences diverge at 10–30 times the mean rate of nuclear genes (*Lynch et al., 2006*; *Allio et al., 2017*). This difference is typically ascribed to a faster underlying mutation rate and limited scope for purifying selection on mitochondrial genes, given uniparental inheritance, negligible recombination, and high ploidy (*Rand, 2001*). At face value, weak selection against mitochondrial mutations might seem to be consistent with the high prevalence of mitochondrial mutations (~1 in 200) (*Elliott et al., 2008*) and diseases (~1 in 5000 births) (*Schaefer et al., 2008*) in human populations. But it is not consistent with the strong signal of purifying selection (*da Fonseca et al., 2008*), evidence of adaptive change (*James et al., 2016*) and codon bias (*Yang and Nielsen, 2008*) in mitochondrial genes, nor with the low transmission rate of severe mitochondrial mutations between generations (*Stewart et al., 2008*; *Fan et al., 2008*; *Hill et al., 2014*). Despite the high rate of sequence divergence, female germline processes apparently facilitate selection against mitochondrial mutations, but the mechanisms are disputed and poorly understood (*Burr et al., 2018*).

Here, we develop computational and evolutionary models that compare three hypotheses of germline mitochondrial inheritance and selection: (i) selection at the individual level, facilitated by mitochondrial bottlenecks; (ii) selection at the cell level through follicular atresia, which weeds out primordial germ cells with high mutation load; and (iii) selection at the organelle level through

selective transfer of mitochondria into oogonia during development. These modes of selection are not mutually exclusive nor the only ones that operate (a fuller examination is provided in the Discussion). The objective here is to clearly delineate the major forces involved and the effectiveness of each in controlling mutation accumulation. Our approach incorporates important factors neglected in earlier work, in particular the input of *de novo* mitochondrial mutations and their segregation over multiple rounds of germ-cell division. This provides a realistic model of mutation, segregation and selection allowing the three hypotheses to be tested against the observed levels of mitochondrial mutation and fitness across a variety of species with an emphasis on the detailed clinical reports in human populations (*Elliott et al., 2008*; *Schaefer et al., 2008*).

The idea that mitochondrial mutations are winnowed through a tight germline bottleneck is pervasive in the literature and has long been held to explain sharp changes in mutation load between generations (*Johnston et al., 2015*; *Floros et al., 2018*; *Stewart and Chinnery, 2015*). The exact size of the bottleneck is unclear, with estimates from mouse (*Wai et al., 2008*; *Cree et al., 2008*; *Cao et al., 2009*) and human studies (*Floros et al., 2018*; *Rebolledo-Jaramillo et al., 2014*; *Guo et al., 2013*; *Li et al., 2016*) spanning two orders of magnitude. Bottlenecks generate variance in mutation loads among the resulting germ cells, and tighter bottlenecks produce greater variance, offering scope for selection against mitochondrial mutations at the level of the individual (*Bergstrom and Pritchard, 1998*; *Roze et al., 2005*; *Hadjivasiliou et al., 2013*). The problem with this line of thinking is that it ignores two other forces. First, gametes are produced through multiple rounds of cell division, leading to repeated rounds of mitochondrial segregation, which in itself generates considerable variance (*Radzvilavicius et al., 2017*). Second, bottlenecks induce greater input of *de novo* mutations as more rounds of mitochondrial replication are required to regenerate the extreme ploidy of mitochondrial DNA in mature oocytes. By applying realistic segregation dynamics and mutational input, we evaluate the impact of these forces on the value of bottleneck size on individual fitness.

Follicular atresia is another force widely considered to be critical in maintaining oocyte quality (*Krakauer and Mira, 1999*; *Chu et al., 2014*; *Haig, 2016*). In humans (*Albamonte et al., 2008*; *Kaipia and Hsueh, 1997*), the number of germ cells declines dramatically in the foetus between mid-gestation (~20 weeks in humans) when there are 7–8 million oocytes, to late gestation when at least two thirds of these are lost, leaving a reserve of 1–2 million at birth (*Townson and Combelles, 2012*). Oocyte loss continues throughout the life of an individual, eventually leading to the depletion of the ovarian pool and loss of reproductive function at menopause (*Suganuma et al., 1993*; *Galimov et al., 2014*; *Cummins, 2004*). Similar loss of female germ cells before sexual maturity is evident in mice and several other animal species (*Nezis et al., 2000*; *Rodrigues et al., 2009*; *Saidapur, 1978*; *Morita and Tilly, 1999*). This attrition has historically been ascribed to cell death during oocyte maturation (*Tilly, 2001*; *Perez et al., 2000*), but more recent findings implicate the apoptotic loss of 'nurse cells' during the genesis of primary oocytes (*Lei and Spradling, 2016*). In either case, differential oocyte loss offers scope for between-cell selection. However, the basis for between-cell selection has long been questioned, on the grounds that it seems unlikely that 70–80% of oocytes have low fitness as a result of mitochondrial mutations (*Perez et al., 2000*). We therefore test whether selection against oocytes with higher loads of mitochondrial mutations during follicular atresia is capable of giving rise to the distribution of mutations observed.

A more recent interpretation of germ-cell loss links it to the formation of the Balbiani body, a prominent feature of the human (*Motta et al., 2000*; *Hertig, 1968*) and mouse (*Lei and Spradling, 2016*; *Kloc et al., 2004*) female germline, as well as a range of other vertebrates and invertebrates, with varying terminology (e.g. fusome, mitochondrial cloud) (*Lei and Spradling, 2016*; *Kloc et al., 2004*; *Tworzydlo et al., 2016*; *Reunov et al., 2019*; *Larkman, 1984*; *Heasman et al., 1984*). In the mouse, proliferating germ cells typically form clusters of five to eight cells that establish cytoplasmic bridges (*Lei and Spradling, 2016*; *Pepling et al., 2007*). It is thought that around half the mitochondria from each nurse cell are streamed into the Balbiani body of the primary oocyte, through an active cytoskeletal process that seems to depend in part on the membrane potential of discrete mitochondria (*Zhou et al., 2010*; *Bilinski et al., 2017*). This offers scope for purifying selection through the preferential exclusion of dysfunctional mitochondria. The remaining nurse cells, now denuded of half their mitochondria, undergo apoptosis (*Lei and Spradling, 2016*). Selective transfer and pooling of mitochondria from interconnected cells may occur in other vertebrate and

invertebrate systems. We consider the consequence of different strengths of selection at the level of mitochondrial function in the production of the Balbiani body.

To systematically distinguish between the predictions of these three different hypotheses, under a range of reasonable parameter values, we use a computational model to evaluate the patterns of mutation load generated over a single generation in each case. We then use an evolutionary model to generate equilibrium levels and compare the predictions to the prevalence of mutations and disease from human studies. Our results show that selection at the organelle level through the pooling of high-quality mitochondria into the Balbiani body is a more potent force than germline bottlenecks and follicular atresia and must play a key role in the maintenance of mitochondrial function in the face of pervasive mutational pressure. This analysis also pleasingly clarifies the longstanding paradox of germ-cell over-proliferation followed by massive loss which is a widely conserved feature of the female germline in animal taxa.

## Results

### Computational model

The computational model follows the distribution of mitochondrial mutations across a single generation, using model parameters derived from human data (*Albamonte et al., 2008*; *Figure 1*). The zygote is assumed to have around half a million copies of mitochondrial DNA (exact number $2^{19}$), which are randomly partitioned to the daughter cells at each cell division. The pattern of segregation is in agreement with recent evidence for actin-mediated mixing of mitochondria within cells during mitosis leading to random segregation (*Moore et al., 2021*). We assume independent segregation of mitochondria with one mtDNA per mitochondrion, and do not consider complications that might arise from the packaging of multiple mtDNA copies per mitochondrion (*Floros et al., 2018*). This assumption is supported by evidence that mitochondrial networks fragment into multiple smaller structures at cell division (*Taguchi et al., 2007*; *Park and Cho, 2012*) that probably contain one or a few mtDNAs.

Mitochondrial replication is not active during early embryo development (*Dumollard et al., 2007*), so the mean mitochondrial number per cell approximately halves with each division (*Figure 1B*). In humans, after 12 cell divisions a random group of 32 cells form the primordial germ cells (PGC) (*Extavour and Akam, 2003*), which in the model corresponds to a mean of 128 mitochondria per PGC. Mitochondrial replication resumes at this point (*Albamonte et al., 2008*; *Dumollard et al., 2007*). Each mtDNA doubles prior to cell division. With probability μ, one of the daughter mitochondria acquires a new deleterious mutation through a copying error. We consider μ in the range $10^{-9}$ to $10^{-8}$ to $10^{-7}$ per base pair per cell division (designated low, standard, and high, respectively), consistent with the range of estimates for the female germline, and assume no back mutations (see Materials and methods). Point mutations during replication are the dominant form of mutation in mtDNA, so we do not consider damage from other sources such as oxidative damage (*Stewart and Larsson, 2014*). Mitotic proliferation of PGCs gives rise to ~8 million oogonia, which are reduced to ~1 million primary oocytes during late gestation (*Figure 1B*; *Albamonte et al., 2008*; *Dumollard et al., 2007*). Proliferation is followed by a quiescent phase during which the mitochondria in primary oocytes are not actively replicated. Mutations accumulate far more slowly during this phase, which persists over decades in humans (*Dumollard et al., 2007*; *Allen and de Paula, 2013*). For simplicity, we assume no mutational input during this period (not marked in *Figure 1B*). At puberty, the primary oocytes mature through clonal amplification of mitochondria back to the extreme ploidy in mature oocytes (~500,000 copies; *Figure 1B*; *Radzvilavicius et al., 2016*). The same copying error mutation rate μ is applied during this process.

We consider three different forms of selection on mitochondria: selection at the level of individuals, cells, or mitochondria. We apply selection at the level of individuals on the zygotic mutation load. Selection at the level of cells or mitochondria is applied during culling at late gestation when primary oocytes are produced. Each of these processes can be captured by modifications of the computational model, allowing easy comparison between them. In order to distinguish between different levels of selection, the model extends earlier work that considered segregational variation of a fixed burden of existing mutations (*Johnston et al., 2015*; *Bergstrom and Pritchard, 1998*; *Roze et al., 2005*; *Hadjivasiliou et al., 2013*) but neglected the input of new mutations during PGC

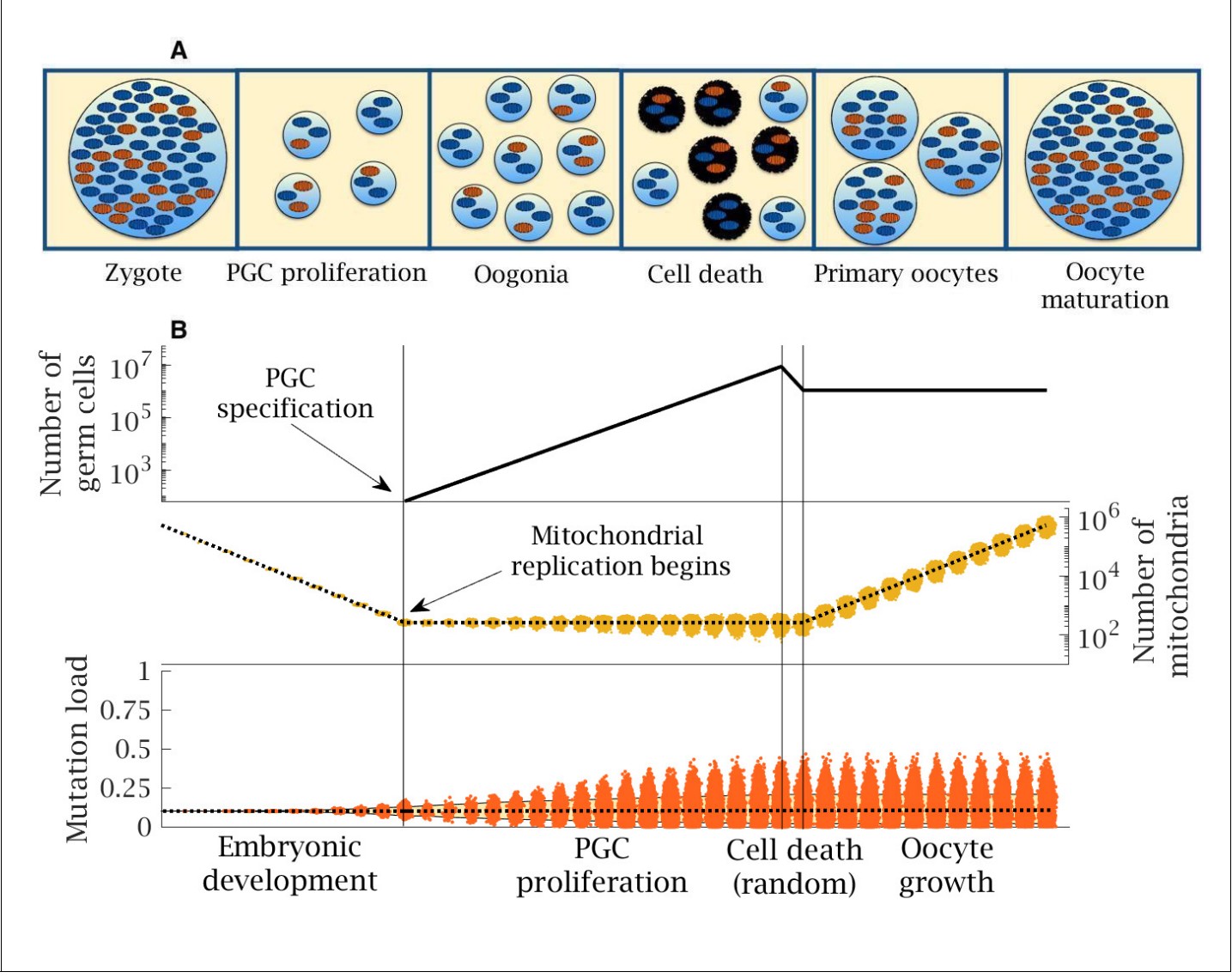

**Figure 1.** Stages in female germline development. (**A**) Timeline of human oocyte development showing the main stages modeled, with wildtype (blue) and mutant mitochondria (orange). (**B**) Numerical simulation of the base model. Top panel: number of germ cells from specification of the 32 primordial germ cells (PGCs) after 12 cell divisions; proliferation to form 8 million oogonia; random cell death reducing to 1 million primary oocytes; quiescent period (not shown) and finally oocyte maturation at puberty. Middle panel: copy number of mitochondria (i.e. mtDNA); from zygote with ~500,000 copies, which are partitioned at cell division during early embryo development until replication begins (first vertical line) during PGC proliferation; copy number is amplified during oocyte maturation back to ~500,000 copies; dotted line shows the mean mitochondria copy number, with the distribution across oocytes shown in yellow. Note, skew reflects the log-scale. Bottom panel: mean (dotted line) and distribution of mutation load through development. The yellow shaded area shows the 90% quantile. Other parameter values $\mu = 10^{-8}$, $m_0 = 0.1$.

proliferation and oocyte maturation, as well as the loss of germ cells during late gestation. The analysis here shows the importance of considering these additional processes governing the population of mitochondria in germline development.

## Germline bottleneck increases variance but introduces more *de novo* mitochondrial mutations

The effect of a bottleneck was assessed in the model by allowing $b$ extra rounds of cell division without mitochondrial replication during early embryonic development (e.g. two extra rounds shown in *Figure 2A*). Each additional cell division leads to an average reduction of $(0.5)^b$ mitochondria in PGCs compared to the base model. For simplicity we then hold mitochondrial numbers at this lower

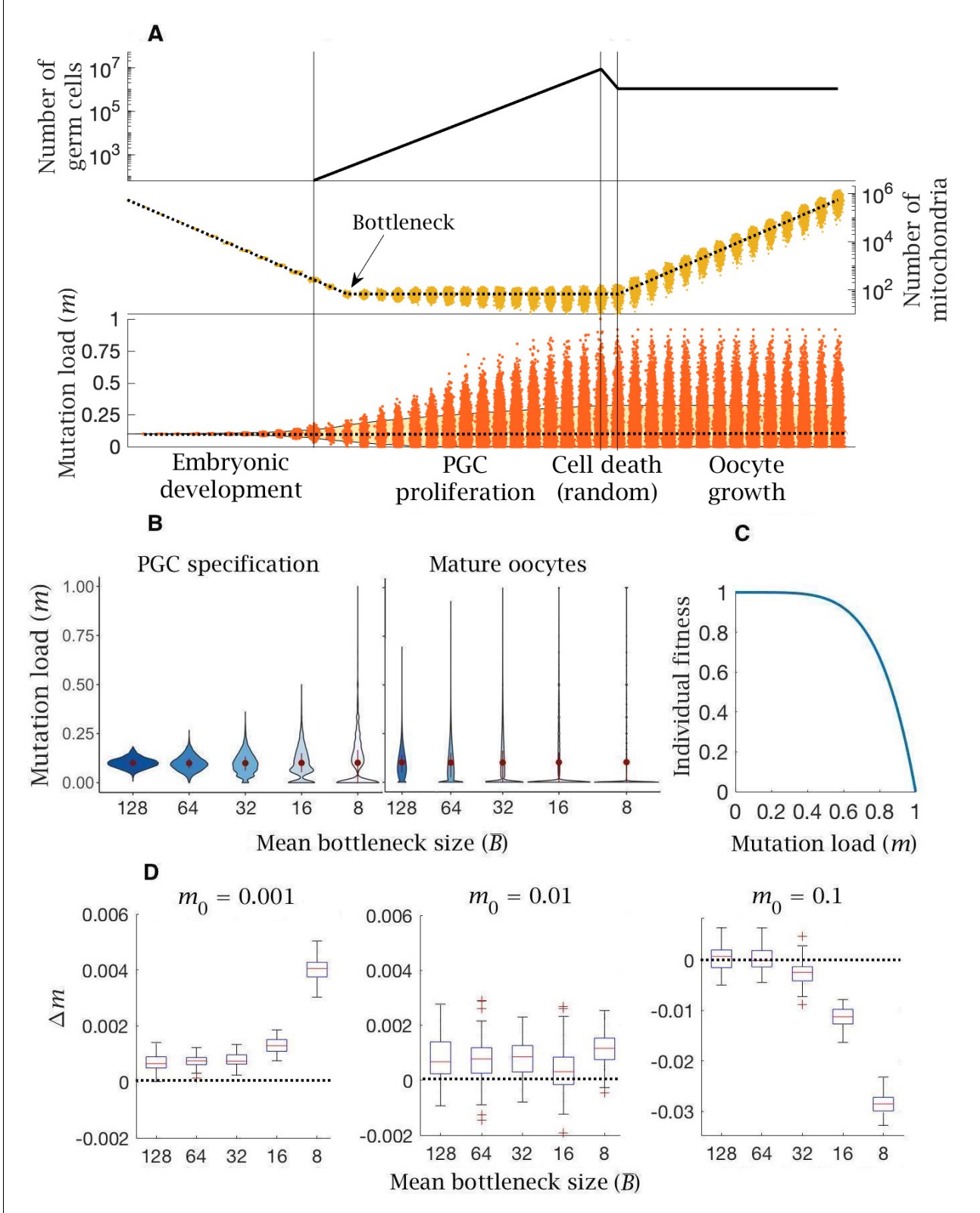

**Figure 2.** Model of germline bottleneck and individual selection. (**A**) A bottleneck with two extra rounds of cell division without replication (cell division 13 and 14; after the first vertical line), reducing mitochondria copy number per PGC (by a quarter on average). Two extra rounds of mitochondrial replication are required to regenerate the copy number in mature oocytes. Compared to the base model (**Figure 1**), mean mutation load (dotted line, bottom panel) is slightly higher and variation in load is substantially greater (yellow shaded area, 90% quantile). Parameter values $\mu = 10^{-8}$, $m_0 = 0.1$. (**B**)
*Figure 2 continued on next page*

*Figure 2 continued*

Violin plots of the distribution of mutations (mean ± SD shown in red) at two developmental stages, PGC specification and mature oocytes, given 5 mean bottleneck sizes ($\bar{B}$) when $m_0 = 0.1$. (C) Strength of selection on individual fitness, with a concave fitness function based on clinical data from mitochondrial diseases. (D) Change in mutation load ($\Delta m$) across a single generation for three initial mutation loads ($m_0$), given 5 mean bottleneck sizes ($\bar{B}$), showing the median (red line) and distribution (box plot IQR with min/max whiskers and outliers).

The online version of this article includes the following figure supplement(s) for figure 2:

**Figure supplement 1.** Bottleneck change in mutation load with different mutation rates.

value through the period of PGC proliferation. This assumption gives greater impact to the bottleneck and is consistent with some views (*Cao et al., 2009*). Tighter bottlenecks at this early developmental stage generate greater segregational variance in mutation load between cells (*Figure 2B*). This increase in variance persists and is enhanced through PGC proliferation to the production of primary oocytes and ultimately in mature oocytes (*Figure 2B*). The bottleneck not only creates a wider spread of mutation number per cell, but also the possibility that cells can be mutation free even when initiated from a zygote that contains significant numbers of mutations (*Figure 2B*). Bottlenecks in themselves do not change the mean mutation load, as they occur before the start of mitochondrial replication (i.e. at PGC specification; *Figure 2B*; *Dumollard et al., 2007*). But oocyte maturation requires the expansion of mitochondrial number back to half a million. Cells starting with lower numbers must therefore undergo more rounds of mitochondrial replication, and hence will accumulate more *de novo* mutations. So, the mean mitochondrial mutation load in mature oocytes increases with tighter bottleneck size, albeit this effect is small with standard mutation rates ($\mu = 10^{-8}$; *Figure 2B*). Nonetheless, the tension between variance and mean determines the overall selective consequence of the bottleneck.

The advantage that the bottleneck brings depends on how selection acts against the mutation load carried by an individual. Based on the observed dependence of mitochondrial diseases on mutation load (*Rossignol et al., 2003*; *Kopinski et al., 2019*; *Wallace and Chalkia, 2013*), in which more serious phenotypes typically manifest only at high mutation loads of >60 % (*Rossignol et al., 2003*; *Kopinski et al., 2019*; *Wallace and Chalkia, 2013*), it is thought that individual fitness is defined by a concave fitness function, indicative of negative epistasis (*Figure 2C*). This assumes that each additional mitochondrial mutation causes a greater reduction in fitness beyond that expected from independent effects. In other words, low mutation loads have a relatively trivial fitness effect, whereas higher mutation loads produce a steeper decline in fitness.

The change in mutation load ($\Delta m$) over a single generation after individual selection was measured against 5 mean bottleneck sizes ($\bar{B} = 128, 64, 32, 16, 8$), for three initial mutation loads ($m_0$) and three mutation rates ($\mu$). The bottleneck shows an ambiguous relationship with fitness, dependent on the inherited mutation load ($m_0$). For the estimated mutation rate ($\mu = 10^{-8}$), there is always an increase in mutation load in individuals who inherit low or medium mutation loads ($m_0 = 0.001, 0.01$; *Figure 2D*). This increase in load ($\Delta m > 0$) becomes more deleterious with a tighter bottleneck (*Figure 2D*). The bottleneck only confers a benefit ($\Delta m < 0$) among individuals who inherit a high mutation load ($m_0 = 0.1$; *Figure 2D*), where the advantage of greater variance outweighs the increase in *de novo* mutation load. If the mutation rate is lower ($\mu = 10^{-9}$), bottlenecks have little effect except when severe, where they again cause an increase in mutation number in individuals with low or medium mutation loads ($m_0 = 0.001, 0.01$; *Figure 2—figure supplement 1A*). In individuals with high mutation load ($m_0 = 0.1$), only tighter bottlenecks ($\bar{B} = 16, 8$) are beneficial (*Figure 2—figure supplement 1A*). If the mutation rate is higher ($\mu = 10^{-7}$) the pattern is more extreme, with the accumulation of *de novo* mutations except in individuals with high inherited mutation loads ($m_0 = 0.1$) at the tightest bottleneck size ($\bar{B} = 8$) (*Figure 2—figure supplement 1B*). In sum: even though bottlenecks generate greater variance, they impose the need for additional rounds of mitochondrial replication during oocyte maturation, resulting in greater *de novo* mutational input. This makes tight bottlenecks advantageous only for rare individuals who inherit high mutation loads, but not for the great majority of the population where the prevalence of mitochondrial mutations is below the limits of detectability, between 0.001 and 0.01 (*Elliott et al., 2008*; *Floros et al., 2018*).

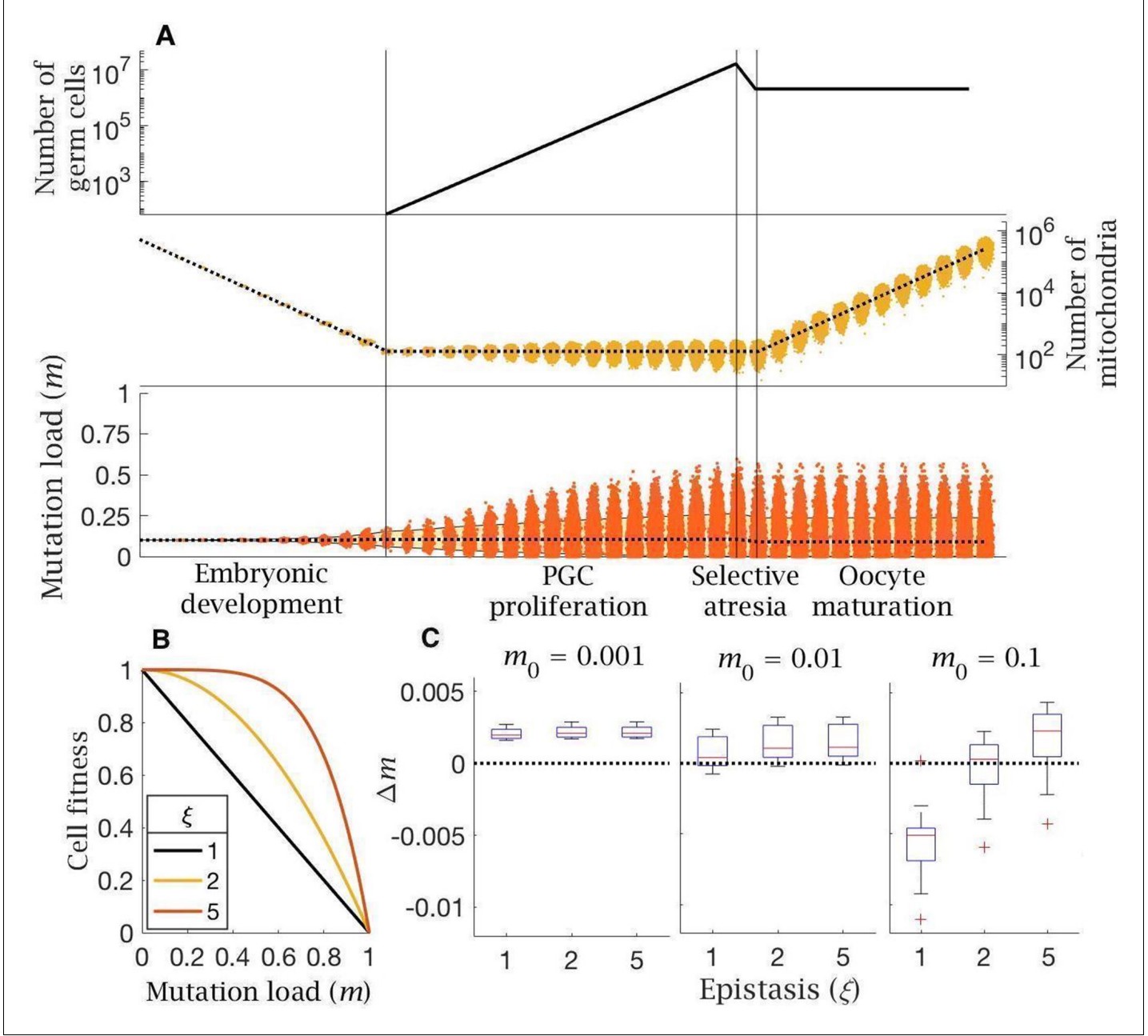

**Figure 3.** Model of follicular atresia and cell selection. (A) After PGC proliferation, follicular atresia occurs through selective apoptosis of oogonia. (B) Cell fitness is assumed to be linear ($\xi = 1$) or follow negative epistasis ($\xi = 2, 5$) in which mutations are more deleterious in combination. (C) Change in mutation load, $\Delta m$, across a single generation after cell selection, at an intermediate mutation rate ($\mu = 10^{-8}$), for individuals with low ($m_0 = 0.001$), medium ($m_0 = 0.01$) and high ($m_0 = 0.1$) initial mutation loads, for variable levels of epistasis (median (red line) and distribution (box plot IQR with min/max whiskers and outliers)).

The online version of this article includes the following figure supplement(s) for figure 3:

**Figure supplement 1.** Follicular atresia and cell selection change in mutation load with different mutation rates.

### Follicular atresia cannot be explained by realistic selection against cells with high mitochondrial mutation loads

In the analysis of bottlenecks above, the culling of ~8 million oogonia to 1 million primary oocytes at the end of PGC proliferation was assumed to be a random process (*Figure 2A*). This loss has a minimal effect on the mean and variance of mitochondrial mutations in germ cells, given the large

numbers involved (and no effect at all when averaged over a population). However, the loss of ~80% of oocytes via follicular atresia during late gestation has long been puzzling and could arguably reflect selection against cells with higher mutation loads.

To analyze follicular atresia, cell-level selection was applied to oogonia at the end of PGC proliferation (*Figure 3A*). PGCs vary in mutation frequency due to both the random segregation of mutants during the multiple cell divisions of proliferation and the chance input of new mutations during mtDNA replication. In principle, we assume that between-cell selection is governed by a negative epistatic fitness function (*Figure 3B*) similar to that thought to apply at the individual level, and vary selection from linear ($\xi = 1$), weak ($\xi = 2$) to strong epistasis ($\xi = 5$). Positive epistasis ($\xi < 1$), whereby a single point mutation produces a steep loss of fitness, but additional mutations have less impact (i.e. mutations are less deleterious in combination), seems biologically improbable, so we do not consider it here.

The effect of cell selection during follicular atresia was calculated as the change in mutation frequency for individuals carrying different mutation loads ($m_0$) over a single generation, given standard values for *de novo* mutations ($\mu = 10^{-8}$) and bottleneck size ($\bar{B} = 128$). Under strong negative epistasis ($\xi = 5$), only the few cells with very high mutation loads (generated by segregation) are eliminated. Cell-level selection does not reduce mutation load, even for individuals with a high initial frequency of mutations ($m_0 = 0.1$; *Figure 3C*). Cell-level selection is more effective with weak epistasis ($\xi = 2$) or linear selection ($\xi = 1$) as this makes cells with lower mutation loads more visible to selection, and has a greater benefit in individuals carrying higher initial mutation loads (*Figure 3C*). However, in individuals who inherit low or medium mutation load ($m_0 = 0.001, 0.01$) cell selection offers a minimal constraint against mutation input. The only case in which cell selection produces a benefit is with high mutation load ($m_0 = 0.1$) under linear selection ($\xi = 1$) (*Figure 3C*). This pattern holds for a lower mutation rate ($\mu = 10^{-9}$; *Figure 3—figure supplement 1A*), while there is no benefit at all at a higher mutation rate ($\mu = 10^{-7}$; *Figure 3—figure supplement 1B*).

## The Balbiani body pools high-quality mitochondria and restricts *de novo* mutation input

An alternative interpretation of atresia in late gestation lies in the formation of the Balbiani body. We model the developmental process giving rise to the Balbiani body by assuming that cysts of 8 oogonia form at the end of PGC proliferation (*Figure 4A*). Cells within a cyst are derived from a common ancestor (i.e. via three consecutive cell divisions). At the eight-cell stage, intercellular bridges form between the oogonia. These allow cytoplasmic transfer of a proportion of mitochondria ($f$) from each cell to join the Balbiani body of the single cell destined to become the primary oocyte (*Figure 4A*). The mitochondria that undergo cytoplasmic transfer are sampled at random (without replacement), with different weights for wildtype ($p_{wt}$) and mutant ($p_{mut}$) mitochondria, until $f$ have moved to the Balbiani body. The oogonia that donate their cytoplasm to the primary oocyte are now defined as nurse cells, and undergo programmed cell death – atresia (*Figure 4A*).

The model shows that two benefits accrue from cytoplasmic transfer. The first benefit of mitochondrial transfer into the Balbiani body is that pooling increases the number of mitochondria in primary oocytes. As the proportion of mitochondria transferred increases towards the estimated rate of $f = 0.5$ (*Lei and Spradling, 2016*), the number of mitochondria in primary oocytes increases four fold. Pooling therefore cuts the number of rounds of replication needed to reach the extreme ploidy required by mature oocytes, which decreases the input of new mutations from replication errors during oocyte maturation. This benefit accrues whatever the initial mutation load, and more dramatically with a higher mutation rate (*Figure 4—figure supplement 1*).

The second benefit arises from selective transfer of mitochondria. Preferential exclusion of mutant mitochondria ($p_{wt} > p_{mut}$), as suggested by experimental evidence (*Lei and Spradling, 2016*; *Bilinski et al., 2017*; *Chen et al., 2020*), lowers the mutation load in primordial oocytes (*Figure 4B*). The difference between $p_{wt}$ and $p_{mut}$ determines the extent to which the mutation load is reduced, with stronger exclusion of mutant mitochondria (lower $p_{mut}$) reducing the number of mutations when the inherited load is medium or high ($m_0 = 0.01, 0.1$), albeit with a negligible effect at low initial mutation load ($m_0 = 0.001$; *Figure 4C*). The same effect is seen with lower and higher mutation rates (*Figure 4—figure supplement 2*). Nurse cells retain a higher fraction of mutant mitochondria but undergo apoptosis, removing mutants from the pool of germ cells, and explaining the need for an

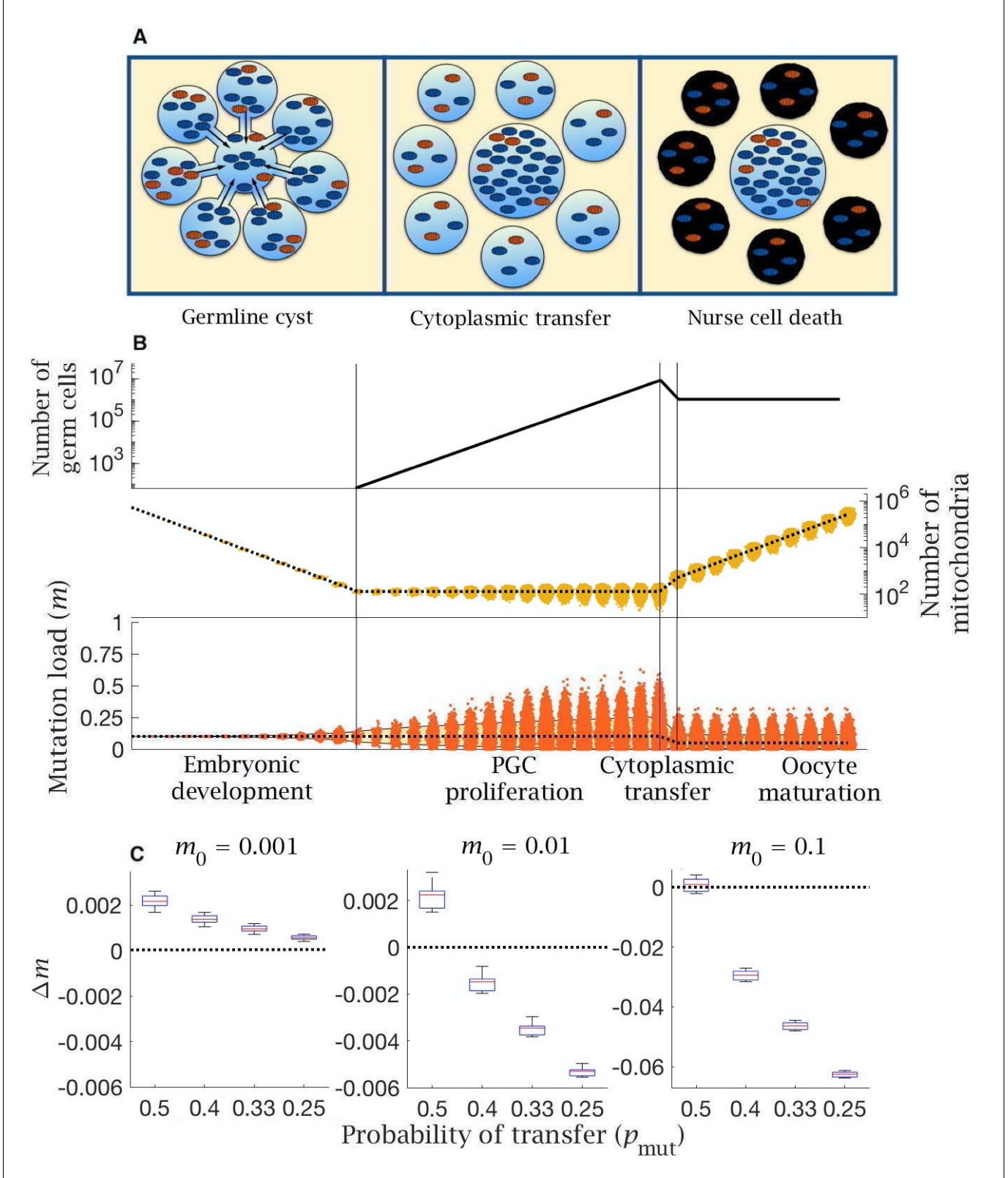

**Figure 4.** Model of cytoplasmic transfer and mitochondria selection. (A) Cytoplasmic bridges form among oogonia in the germline cyst, leading to selective transfer of wild-type mitochondria (blue) to the primary oocyte, leaving mutant mitochondria (red) in nurse cells that then undergo apoptosis. (B) Cytoplasmic transfer which selectively pools $f = 50\%$ of mtDNA from eight germline cyst cells into a single primary oocyte causes a large increase in the number of mitochondria (middle panel) and a large reduction in the mean (dotted line, bottom panel) and distribution of mutation load (yellow

*Figure 4 continued on next page*

*Figure 4 continued*

shaded area shows the 90% quantile, bottom panel), which persists during oocyte maturation. Pooling of mtDNA requires two fewer rounds of mtDNA replication to regenerate copy number in mature oocytes. Parameter values $\mu = 10^{-8}$, $m_0 = 0.1$. (C) Change in mutation load ($\Delta m$) across a single generation (median (red line) and distribution (box plot IQR with min/max whiskers and outliers)), for individuals with low ($m_0 = 0.001$), medium ($m_0 = 0.01$), and high ($m_0 = 0.1$) initial mutation loads, with variable strengths of selective transfer ($p_{mut}$). Parameter value $\mu = 10^{-8}$, $p_{wt} = 0.5$.

The online version of this article includes the following figure supplement(s) for figure 4:

**Figure supplement 1.** Cytoplasmic transfer and mitochondria selection change in mutation load with different mutation rates and proportion of transferred mitochondria (*f*).

**Figure supplement 2.** Cytoplasmic transfer and mitochondria selection change in mutation load with different mutation rates and probability of mutant transfer (*p_{mut}*).

extreme loss of germ cells during late gestation. This effect acts in concert with pooling leading to a reduction in both the mean and variance of mitochondria mutation load in the cells destined to develop into mature oocytes. (*Figure 4B*).

## Evolutionary model

The computational model discussed above gives an indication of the effectiveness of selection at the level of individuals, cells or mitochondria in eliminating mitochondrial mutations across a single generation. To address the long-term balance of mutation accumulation versus selection over many generations, we developed an evolutionary model. This assesses the effectiveness of the three representations of germline development in explaining the observed prevalence of mitochondrial mutation load and disease in human populations (see Materials and methods). This evolutionary model evaluates long-term evolutionary change in an infinite population with non-overlapping generations and is implemented using a number of approximations, which greatly reduce the model complexity (see Materials and methods).

By iterating the patterns of germline inheritance and selection, the equilibrium mutation distribution was calculated across a range of mutation rates and bottleneck sizes. The accuracy of the three models was then assessed as the likelihood of reproducing the observed levels of mitochondrial mutations in the human population (*Figure 5*). Specifically, we used estimated values of 1/5000 for mitochondrial disease (>60% mutant), 1/200 for carriers of mitochondrial mutants (2–60% mutant) and hence 99.5% of individuals are 'mutation free' (i.e. carry <2% mutants, the threshold for detection in these estimates of mutation frequency [*Elliott et al., 2008*; *Schaefer et al., 2008*]). Recent deep-sequencing estimates using a mutation detection threshold of >1% (*Floros et al., 2018*), show that a minor allele frequency of 1–2% is relatively common in selected human PGCs, but this does not alter earlier population-level estimates of the proportion of carriers not suffering from overt mitochondrial disease, defined as a 2–60% mutation load used here.

Likelihood heatmaps confirm that selection at the level of individuals or cells alone do not readily approximate the clinical data whatever the bottleneck size (*Figure 5A–B*). Only at a mutation rate $\mu < 0.5 \times 10^{-8}$ do these forms of selection explain the observed mutation load and disease frequency in humans at high likelihood, especially when using tighter bottlenecks (*Figure 5A*). These limitations do not apply to the preferential transfer of wildtype mitochondria into the Balbiani body (*Figure 5C–D*). Even intermediate levels of selection against the transfer of mutant mitochondria into the Balbiani body ($p_{mut} = 0.33$, $p_{wt} = 0.67$) generates a high log-likelihood of reproducing the clinical data at the standard mutation rates ($\mu = 10^{-8}$) and bottleneck sizes (> 100 mitochondria per cell) (*Figure 3C*). Stronger selection on transfer probabilities ($p_{mut} = 0.25$, $p_{wt} = 0.75$) can account for the clinical pattern under a wide range of bottleneck sizes and mutation rates (*Figure 5D*).

## Discussion

How selection operates on mitochondria has long been controversial. At the heart of this problem is the paradox that mtDNA accumulates mutations faster than nuclear genes, yet there is evidence that mtDNA is under strong purifying selection. Mitochondrial mutations accumulate through Muller's ratchet, as mtDNA is exclusively maternally inherited, and does not undergo recombination through meiosis (*Rand, 2001*). In addition, mitochondrial genes are highly polyploid, which obscures the relationship between genotype and phenotype, hindering the effectiveness of selection on

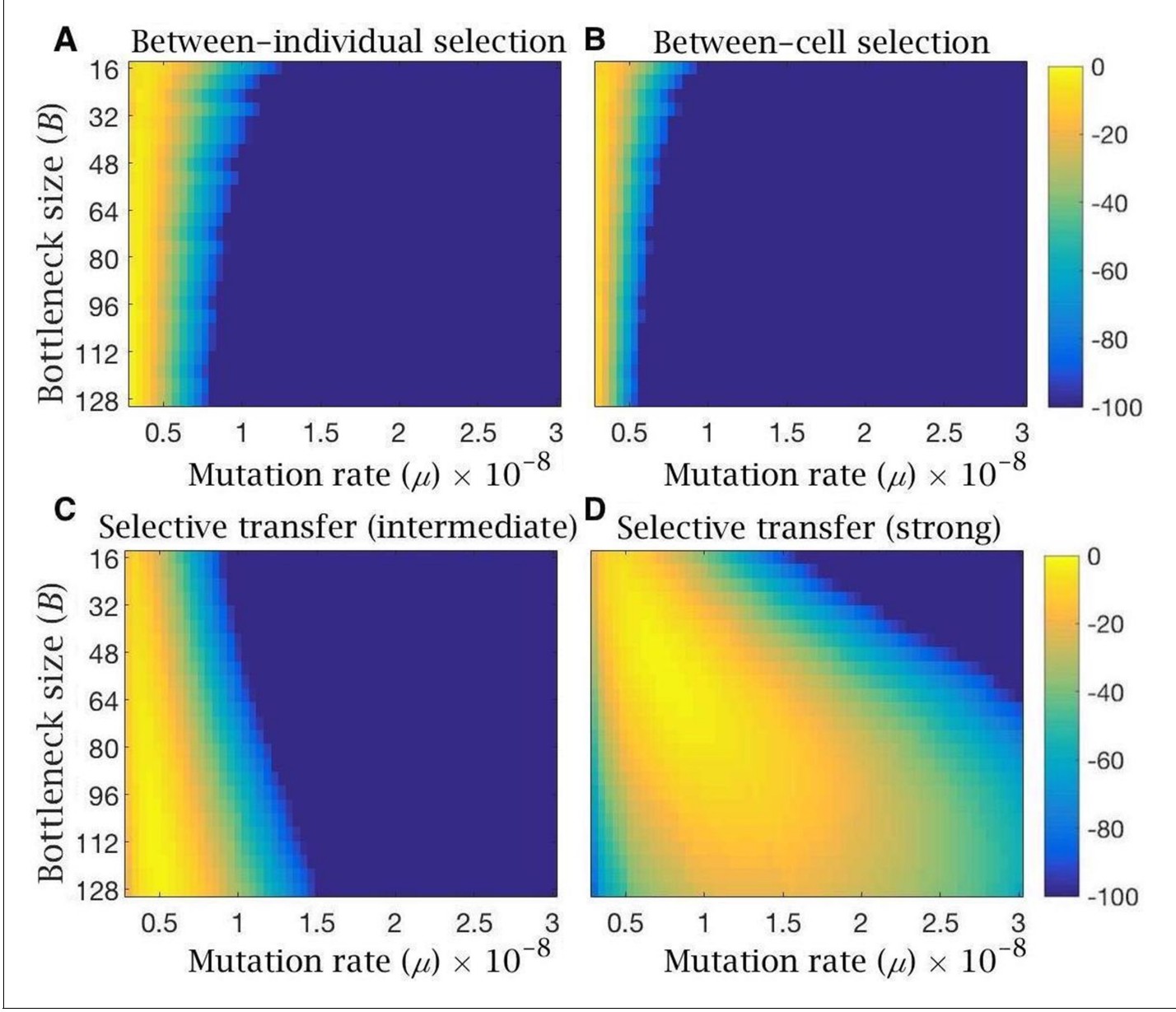

**Figure 5.** Log-likelihood of the models reproducing clinical data of mitochondria mutation load and disease frequency. Heatmaps showing log-likelihood of reproducing the observed mutation load and disease frequency in humans, for equilibrium conditions under the evolutionary model with (A) bottleneck and selection on individuals, (B) follicular atresia and selection on cells ($\xi = 5$), (C) cytoplasmic transfer with intermediate ($p_{mut} = 0.33$, $p_{wt} = 0.67, f = 0.5$) or (D) strong ($p_{mut} = 0.25$, $p_{wt} = 0.75, f = 0.5$) selective transfer of wild-type mitochondria. Yellow depicts high likelihood; blue, low likelihood. All models are shown for variable bottleneck size (the minimum mitochondria population size at which replication commences) and variable mutation rates.

individuals. Despite these constraints, deleterious mitochondrial mutations seem to be eliminated effectively (*da Fonseca et al., 2008*; *James et al., 2016*; *Yang and Nielsen, 2008*; *Stewart et al., 2008*; *Fan et al., 2008*; *Hill et al., 2014*), facilitated by female germline processes that have long been mysterious. These include: the excess proliferation of primordial germ cells (PGCs) (*Pepling, 2006*), the germline mitochondrial bottleneck (when mitochondrial numbers are reduced to a disputed minimum in PGCs) (*Johnston et al., 2015*; *Floros et al., 2018*; *Stewart and Chinnery, 2015*), the formation of the Balbiani body in primary oocytes (*Lei and Spradling, 2016*; *Bilinski et al., 2017*), the atretic loss of 70–80% of germ cells during late gestation (*Townson and Combelles, 2012*; *Tilly, 2001*), the extended oocyte quiescence until puberty or later (during which

time mitochondrial activity and replication is suppressed) (*Allen and de Paula, 2013*; *de Paula et al., 2013*) and the generation of around half a million copies of mtDNA in mature oocytes (*Radzvilavicius et al., 2016*). The key question is how do these processes facilitate the maintenance of mitochondrial quality over generations?

In this study, we introduced a computational model that considers these germline processes from the perspective of mitochondrial proliferation, segregation, and selection, using realistic estimates of parameter values, drawn from the human literature (*Albamonte et al., 2008*, *Dumollard et al., 2007*). Most work to date (*Johnston et al., 2015*, *Stewart and Chinnery, 2015*; *Wai et al., 2008*; *Cree et al., 2008*; *Cao et al., 2009*) has focused on the mitochondrial bottleneck as a means of generating variation in mitochondrial content between oocytes and by extension zygotes (*Figure 2B*), furnishing the opportunity for selection to act on individuals in the following generation. These studies have been unable to reconcile serious differences in experimental estimates of mitochondrial numbers during PGC proliferation, inciting inconclusive debates over the tightness of the bottleneck (*Johnston et al., 2015*; *Stewart and Chinnery, 2015*; *Wai et al., 2008*; *Cree et al., 2008*; *Cao et al., 2009*). More significantly, this earlier work neglects an important germline feature, the introduction of *de novo* mitochondrial mutations produced by copying errors (*Arbeithuber et al., 2020*) rather than damage by reactive oxygen species (*Stewart and Larsson, 2014*, *Trifunovic et al., 2005*). These accumulate during PGC proliferation and, equally importantly, during the mass-production of mtDNAs in the mature oocyte. Tighter bottlenecks are disadvantageous as they impose the need for more rounds of mitochondrial replication which means a greater input of *de novo* mutations. Our modelling shows that for most individuals the mean mutation load shows little meaningful change (*Figure 2D*), regardless of whether the mutation rate is set low or high (*Figure 2—figure supplement 1*), and in fact increases with tighter bottleneck size (*Figure 2D*). Most individuals have low mutation loads (~99.5% in human populations [*Elliott et al., 2008*; *Schaefer et al., 2008*]), and for them, the normal process of repeated segregation during cell division generates sufficient variance in itself. Any marginal increase in variance caused by bottlenecks is more than offset by increased mutational input. Tighter bottlenecks only benefit individuals who already carry high mutation loads (i.e. $m_0 \geq 0.1$, *Figure 2D*). For them, there is benefit in further reductions in bottleneck size as this increases the fraction of mature oocytes with significantly reduced mutation load (*Figure 2D*). In the modeling, we assumed that the bottleneck size was maintained across the period of PGC proliferation. Some studies have found that from a low number in early development, copy number increases 5-10 fold to production of the oogonia (*Wai et al., 2008*; *Cree et al., 2008*). This would lessen the effect of the bottleneck in general as it would have less effect on segregation.

These results show that the popular idea that a germline mitochondrial bottleneck facilitates selection against mitochondrial mutations is misconstrued. The value of a bottleneck depends on the unforeseen trade-off between increasing genetic variance and mutation input. In fact, the reduction in mitochondrial copy numbers from zygote to primordial germ cells should be thought of as the reestablishment of a typical copy number at the start of cellular differentiation, which commences after multiple cell divisions *without* mtDNA replication. What counts as a bottleneck are the 'extra' rounds of cell division reducing mitochondrial number below the 'normal' number, and the incremental increase in variance this induces. Most critically, the bottleneck needs to be understood in relation to oogamy, the massively exaggerated mitochondrial content of the female gamete. This is a characteristic of metazoan gametogenesis (*Radzvilavicius et al., 2016*). Previous work has shown it is beneficial in animals with mutually interdependent organ systems (*Radzvilavicius et al., 2016*). The extreme ploidy in the zygote allows early rounds of cell division to occur without mitochondrial replication, and hence without *de novo* mutational input. These initial cell divisions generate little between-cell differences, as segregational variance is weak when numbers are high and mitochondria segregate randomly during mitosis (*Moore et al., 2021*) (e.g. *Figure 1B* before PGC specification). So at the point of cellular differentiation (~12 cell divisions), there is homogeneity in the mutation load among the different organ systems and no one system is likely to fail, which would massively lower the fitness of the whole organism (*Radzvilavicius et al., 2016*). This contrasts with organisms that have modular growth, such as plants and morphologically simple metazoa (sponges, corals, placozoa), which neither sequester a recognizable germline distinct from the stem-cell lineage early in development (although recent work challenges this view, *Lanfear, 2018*), nor have oocytes

with massively expanded mitochondrial numbers (*Extavour and Akam, 2003*; *Radzvilavicius et al., 2016*; *Extavour, 2007*; *Blackstone and Jasker, 2003*).

Follicular atresia is another female germline feature examined in our modelling, in which there is over-proliferation of PGCs followed by ~80% loss early in development, before oocyte maturation (*Townson and Combelles, 2012*; *Tilly, 2001*). This massive reduction in germ cell number has long been enigmatic. It is unlikely to be random, yet does not obviously serve a selective function, as it seems unlikely that such a high proportion of germ cells could have low fitness (*Krakauer and Mira, 1999*; *Chu et al., 2014*; *Haig, 2016*). The model confirms this intuition. Selection among PGCs at the end of the period of proliferation has little effect in significantly reducing mutation load (*Figure 3C*). Assuming a concave fitness function (*Figure 3B*), which seems reasonable by extension from the severity of mitochondrial diseases (*Rossignol et al., 2003*; *Wallace and Chalkia, 2013*), between-cell selection is ineffective, as it only eliminates PGCs with very high mutational numbers. This has little effect in constraining the burgeoning of lower mutation loads. Linear selection does better, even if it seems unrealistic, as it will act against a broader range of mutational states. But as with bottlenecks, it is only beneficial in individuals already carrying significant mutation loads (i. e. $m_0 \geq 0.1$, *Figure 3C*). We conclude that cell-level selection produces little measurable reduction in mutation load and so is unlikely to be responsible for follicular atresia.

A more recent explanation of PGC loss relates to the formation of the Balbiani body in primary oocytes (*Lei and Spradling, 2016*; *Kloc et al., 2004*). In many metazoa, including clams (*Reunov et al., 2019*), insects (*Tworzydlo et al., 2016*; *Cox and Spradling, 2003*), mice (*Pepling et al., 2007*) and probably humans (*Kloc et al., 2004*), the over-proliferation of PGCs culminates in their organization into germline cysts of multiple oogonia connected by cytoplasmic bridges (*Lei and Spradling, 2016*; *Pepling et al., 2007*; *Cox and Spradling, 2003*). These connections are thought to allow the transfer of mitochondria and other cytoplasmic constituents by active attachment to microtubules, into what becomes the primary oocyte (*Lei and Spradling, 2016*). The surrounding oogonia that transferred their mitochondria, now termed nurse cells, die by apoptosis (*Lei and Spradling, 2016*). The plethora of terms should not mask the key point that nurse cell death accounts for a considerable fraction of the germ cell loss usually ascribed to follicular atresia. We modeled selective mitochondrial transfer into the Balbiani body, perhaps in part reflecting membrane potential (*Tworzydlo et al., 2016*; *Bilinski et al., 2017*). This achieves two complementary benefits: it purges mutations and pools high-quality mitochondria in a single cell. If the germline cyst is composed of eight cells that contribute half of their mitochondria to the Balbiani body then the primary oocyte gains four times as many mitochondria which have passed through quality control. This also cuts the need for additional rounds of mtDNA copying, and so reducing the input of *de novo* mutations. Selective transfer and pooling lowers the mutation load across a wide range of mutation rates and inherited loads (*Figure 4C*, *Figure 4—figure supplements 1–2*). This process differs from mitophagy, the main route used in somatic cells for maintaining mitochondrial quality (*Twig et al., 2008*; *Kim et al., 2007*), as it not only removes mutant mitochondria, but crucially also increases mitochondrial numbers, a key requirement for prospective gametes. The requirement for pooling of mitochondria to lower the mutation load from copying errors also aligns with experimental observations of active spindle-associated mitochondrial migration to the generative oocyte in the formation of polar bodies during meiosis I of oogenesis (*Dalton and Carroll, 2013*). We predict that selection for mitochondrial quality occurs during this process (i.e. polar bodies retain mutant mitochondria) but have not dealt with that explicitly in the model.

These insights depend in part on the parameter values used in the modeling, many of which are uncertain. We have examined variation around the most representative values drawn from the literature (*Allio et al., 2017*; *Floros et al., 2018*; *Sigurğardóttir et al., 2000*; *Stewart and Chinnery, 2015*), and aimed to be conservative wherever possible. We considered mutation rates across two orders of magnitude, around $10^{-8}$ per bp as the standard (*Sigurğardóttir et al., 2000*) and a similar range of bottleneck sizes ($\bar{B} = 8 - 128$). Strong selective pooling of mitochondria into the Balbiani body predicts the observed prevalence of mitochondrial mutations and diseases in human populations (*Elliott et al., 2008*; *Schaefer et al., 2008*) under a wide range of mutation rates and bottleneck sizes (*Figure 5*). Selection at the level of individuals or cells are much more constrained explanations, although we do not rule out some role for these processes (*Figure 5*). In general, higher mutation rates ($10^{-7}$ per base pair) strengthen the conclusions discussed here (*Figure 2—*

*figure supplement 1*), whereas the lowest mutation rates are more commensurate with weaker forms of evolutionary constraint generated by selection on individuals or cells. Plainly, weaker selection approximates best to clinical data when the mutation input tends toward zero (*Figure 5*). However, such low mutation rates are not consistent with the 10-30-fold faster evolution rates of mtDNA compared with nuclear genes (*Lynch et al., 2006*; *Allio et al., 2017*), or with the strong signatures of purifying (*da Fonseca et al., 2008*) and adaptive (*James et al., 2016*) selection on mitochondrial genes. In the modelling, we ignored the contribution of oxidative damage caused by reactive oxygen species. While this source of mutation is likely low compared with copying errors (*Stewart and Larsson, 2014*; *Arbeithuber et al., 2020*), oxidative mutations may accumulate over female reproductive lifespans (*Trifunovic et al., 2005*), perhaps contributing to the timing of the menopause (*Shoubridge and Wai, 2007*). As primary oocytes contain ~6000 mitochondria (*Shoubridge and Wai, 2007*), expansion up to ~500,000 copies in the mature oocyte will amplify any mutations acquired during oocyte arrest at prophase I, potentially over decades (*Arbeithuber et al., 2020*). The metabolic quiescence of oocytes can best be understood in light of the need to repress mitochondrial mutation accumulation during the extended period before reproduction (*Allen and de Paula, 2013*; *de Paula et al., 2013*).

We have addressed here a simple paradox at the heart of mitochondrial inheritance. Like Gibbon's *Decline and Fall of the Roman Empire*, mitochondrial DNA is often portrayed as being in continuous and implacable decline through Muller's ratchet (*Rand, 2001*) yet like the Empire, which endured for another millennium, mitochondrial DNA has persisted and has been at the heart of eukaryotic cell function for over a billion years (*Lane, 2005*). Strong evidence for purifying and adaptive selection implies that the female germline facilitates selection for mitochondrial quality, but the mechanisms have remained elusive. We have modeled segregation and selection of mitochondrial DNA at each stage of germline development, and shown that direct selection for mitochondrial function during transfer into the Balbiani body is the most likely explanation of the observed prevalence of mitochondrial mutations and diseases in human populations. More remarkably, this mitochondria-centric model elucidates the complexities of the female germline. It explains why mature oocytes are crammed with mitochondria (*Radzvilavicius et al., 2016*), whereas sperm mitochondria are typically destroyed, giving rise to two sexes (*Radzvilavicius et al., 2017*); why germ cells over-proliferate during early germline development; why oogonia organize themselves into germline cysts, forming the Balbiani body; why the majority of germ cells then perish by apoptosis as nurse cells; why primary oocytes enter metabolic quiescence, sometimes for decades; and even why polar bodies channel most of their mitochondria into a single mature oocyte. The need for mitochondrial quality extends to somatic cells, as mitochondria activity is crucial to cellular, tissue, and organ functioning in the adult organism (*Pereira et al., 2021*; *Carelli et al., 2015*; *Diot et al., 2016*). Some of the approaches we have adopted here need to be applied to development and whether specific processes have evolved to maintain mitochondria where their function is more critically related to somatic fitness (*Radzvilavicius et al., 2016*; *Buss, 1987*). Most fundamentally, this perspective challenges the claim that complex multicellularity requires passage through a single-celled, haploid stage to constrain the emergence of lower-level, selfish genetic elements (*Buss, 1987*; *Maynard Smith and Szathmáry, 1995*). This is true for nuclear genes in oocytes, whose quality is maintained by sexual exchange and recombination (*Maynard Smith and Szathmáry, 1995*), but is not the case for mitochondria, which are generally transmitted uniparentally, without sexual exchange or recombination. In animals, the oocyte cytoplasm is not derived from a single cell, but instead requires the selective pooling of mitochondrial DNA from clusters of progenitor cells, which together generate high-quality mitochondria at extreme ploidy in mature gametes.

## Materials and methods

### Computational model

#### Initial conditions

We use a computational model implemented in MATLAB (RRID:SCR_001622) to follow the distribution of mitochondrial mutations in the female germline over a single generation from zygote to a new set of mature oocytes, as set out in the developmental history given in the main text (*Figure 1A*). The initial state of the system is a zygote containing $M_0 = 2^{19} = 524,288$ copies of

mtDNA (*Radzvilavicius et al., 2016*; *Reynier et al., 2001*), of which $m_0$ carry a deleterious mutation. Three specific models are considered: bottleneck, follicular atresia and cytoplasmic transfer. A list of terms and parameter values is given in *Table 1*, which also apply in the evolutionary model considered below.

## Early embryonic development

During early embryonic development, there is no mtDNA replication. The number of cells doubles at each time step. The existing population of mutant and wildtype mtDNA undergoes random segregation into daughter cells according to a binomial distribution – each mtDNA copy has a 50% probability of being assigned to either daughter cell. During this process, the average number of mtDNA copies per cell halves at each time step. There is no mutational input, as we only consider mutations that arise due to replication errors.

## PGC proliferation, oogonia cell death, and oocyte maturation

The early embryonic period lasts for the first 12 cell divisions. A group of 32 cells is selected at random to form the primordial germ cells (PGC). The PGCs then undergo proliferation for a further 18 rounds of cell division, until the maximum number of germ cells is reached, $N_{max} = 32 \times 2^{18} = 8,388,608$. This value is close to the average reported in the literature (*Albamonte et al., 2008*).

mtDNA replication resumes after cell division 12, at the point of PGC determination. At this point, cells have an average of 128 mtDNA copies. At each following time step, the number of mtDNA copies doubles prior to random segregation into daughter cells. This means that the average number of mtDNA copies per cell is kept constant. New mutations are introduced as errors in mitochondrial replication. During the replication process, the new replica of each wild-type mtDNA copy has a probability of mutation μ per bp. The genome wide mutation rate $U = g \times \mu$ is calculated as genome size ($g = 16,569$ bp, *Palca, 1990*) multiplied by μ. This estimate assumes each site contributes equally to selective effects and ignores many subtleties relating to mutation probability and within-cell maintenance processes, but should give a reasonable order of magnitude gauge of the target size of mutational input per cell division. Given $n$ wildtype and $m$ mutant mtDNAs, the number of new mutants $\Delta m$ resulting from replication errors is obtained by sampling at random from a binomial distribution with $n$ trials with probability $U$. After replication and mutation and prior to segregation the total number of wildtype and mutant mtDNAs is $2n - \Delta m$ and $2m + \Delta m$, respectively. Back mutation to wildtype is not permitted.

At the end of PGC proliferation, the $N_{max}$ oogonia undergo random cell death, leaving $N_{max}/8 = 1,048,576$ primary oocytes. This is achieved by sampling the surviving cells at random with uniform weights (i.e. every cell has an equal probability of survival). The primary oocytes do not undergo further cell division or mitochondrial replication during the quiescent period (this is not explicitly modeled). At puberty, oocyte maturation commences. The number of mitochondria per cell is brought back to the original value $M_0 = 2^{19}$ through 12 rounds of replication without cell division. We assume that the number of mtDNA copies doubles at each time step. This introduces new

**Table 1.** Parameter and variable symbols and values.

| | |
|---|---|
| Maximum number of germ cells | $N_{max} = 8,388,608$ |
| mtDNA number in mature oocytes | $M_0 = 2^{19} = 524,288$ |
| Minimum mtDNA ploidy | $B$ |
| Final number of germ cells | $N_{max}/8 = 1,048,576$ |
| Initial mutation load | $m_0$ |
| Mutation rate per bp per cell division | $\mu$ |
| Strength of epistatic interactions | $\xi$ |
| Transfer probability of mutant mtDNA | $p_{mut}$ |
| Transfer probability of wildtype mtDNA | $p_{wt}$ |
| Human mitochondrial genome size | $g = 16,569$ bp (*Palca, 1990*) |

deleterious mutations, which again are randomly drawn from a binomial distribution (as described above).

## Specific models of selection

We consider three specific models in the main text with modifications to the base model described above.

The first model adds a bottleneck stage at the time of PGC determination (*Figure 2*). As before, 32 cells are selected at cell division 12 to form the PGCs. These go through $b$ extra rounds of cell division without mtDNA replication. This reduces the mean number of mtDNA copies per cell to $\bar{B} = 128 * (0.5)^b$. The mtDNA replication commences at cell division 14. The PGCs then proliferate as before to produce oogonia that undergo random cell death to produce primary oocytes. The primary oocytes have a reduced number of mtDNA copies, and so must undergo $12 + b$ extra rounds of mtDNA replication in order to regain the original value $M_0$ mitochondria in mature oocytes. Note that this is an extreme model of the bottleneck, where mtDNA copy number is kept low throughout the period of PGC proliferation, and so maximizes the benefit derived from the increase in segregational variation caused by the bottleneck.

For the model of the bottleneck, we allow selection dependent on individual fitness in relation to their mutation load $m$ among mature oocytes, according to the fitness function $f(m) = 1 - \left(\frac{m}{M}\right)^5$ (*Figure 2C*). The concave shape of this function accounts for the fact that mitochondrial mutations typically have a detrimental effect on individual fitness only for loads >60%. Changes to the power exponent make little qualitative difference to the outcome of this model (data not shown).

A second model considers non-random death during the cull of oogonia as these cells transition to being primary follicles (*Figure 3*). Selection in this case is applied at the cell level. Cell fitness is expressed as $f(m) = 1 - \left(\frac{m}{M}\right)^\xi$, where $m$ is the number of mutant mitochondria. The parameter $\xi$ determines the strength of epistatic interactions (*Figure 3B*). As in other models, the number of cells is reduced from $N_{max} = 8,388,608$ to $N_{max}/8 = 1,048,576$. This is achieved by sampling without replacement the surviving cells at random, with weights proportional to cell fitness (i.e., every cell has a probability of survival proportional to its fitness).

The third model considers that the oogonia are organized in cysts of eight cells each. These are the descendants of a single cell (i.e. three cell divisions prior). One cell is randomly designated as the primary oocyte using the MATLAB function randsample. The Balbiani body of the primary oocyte contains a proportion $f$ of the mtDNA copies of all cells in the cyst. The mitochondria that join the Balbiani body are sampled at random without replacement from each cell with different weights for wildtype ($p_{wt}$) and mutant ($p_{mut}$). After mitochondrial transfer to the Balbiani body, nurse cells undergo apoptosis (i.e. all cells except the one designated as the primary oocyte), reducing the total number of oocytes to $N_{max}/8 = 1,048,576$.

## Evolutionary model

In order to calculate the equilibrium distribution of a population undergoing the developmental dynamics mentioned in the previous section, we develop an analytical model for the distribution of mitochondrial mutations in an infinite population, with non-overlapping generations. As it was not possible to find an analytical solution, we solved the equations through numerical iterations. The system converges to a unique equilibrium state, independent of the initial conditions.

The state of the system is described by the vector $p(t) = \{p_0(t), \ldots, p_{M(t)}(t)\}$, where $M(t)$ is the number of mtDNA copies per cell at time $t$. The elements $p_m(t)$ are the frequency of mutation load $m(t)/M(t)$ at time $t$. The evolution of the system is determined by a set of transition matrices whose elements are the transition probabilities between states. To avoid unnecessary complexity in the evolutionary model, we assume that fluctuations in mitochondrial number per cell due to segregation are negligible (i.e. in contrast to the computational model which allows binomial segregation at each division). Therefore, the mtDNA number per cell is constant across the whole population of cells at every time step. That is, during early embryonic development (when there is no mtDNA replication), after $t$ cell divisions, the total number of mitochondria per cell is $M^{(t)} = 2^{-t}M_0$. Then, during PGC proliferation, the total number of mitochondria per cell is constant. Finally, during oocyte maturation, the number of mitochondria per cell exactly doubles with each mtDNA replication cycle. To

aid in calculations, we also set the initial number of mtDNA copies to be proportional to the bottleneck size, that is, $M_0 = 2^{12} \times B$. As the mtDNA number per cell halves at each cell division during early embryonic development, setting $M_0$ this way allows the mtDNA number to remain an integer. This is important for the modeling procedure, because the dimension of the transition matrixes (which is determined by the mtDNA number) must be an integer.

## Early embryonic development

During early embryonic development, when mitochondrial replication is not active, changes in frequency arise purely from the process of segregation. Let $W^{(t)}$ be a $M^{(t)} + 1 \times M^{(t)} + 1$ square matrix, whose elements $W_{mn}$ represent the transition probabilities from a state with $m$ to a state with $n$ mutants:

$$W_{mn}^{(t)} = \binom{m}{n}\binom{M^{(t-1)} - m}{M^{(t)} - n} \Big/ \binom{M^{(t-1)}}{m^{(t)}} \tag{1}$$

These matrix elements model the probability of transitioning from a state with $m$ mutants and $M - m$ wildtype to a state with $n$ mutants and $M - n$ wild type via the segregation of $2M$ mitochondria into two daughter cells with $M$ mitochondria each.

After $t$ cell divisions, the average number of mutants per cell is $\bar{m} = 2^{-t}m_0$, and the variance is $Var(t) = \frac{1}{4}[Var(t-1) + 2^{-t}m_0]$. The state of the system is updated as $\overrightarrow{p}^{(1)} = \left(\prod_t W^{(t)}\right) \times \overrightarrow{p}^{(0)}$.

## PGC proliferation, oogonia cell death, and oocyte maturation

During PGC proliferation, new mutations are introduced at a rate $U = \mu \times g$. The transition coefficient $Q_{mn}$ from a state with $m$ to a state with $n$ mutants results from the combined effects of replication, mutation, and segregation:

$$\begin{aligned} Q_{mn} &= \sum_k \binom{M - n}{k - n} U^{k-n}(1 - U)^{M-k} \binom{k}{m}\binom{M - k}{M - m} \Big/ \binom{2M}{M} \\ &= \sum_k \binom{M - n}{k - n} U^{k-n}(1 - U)^{M-k} a_{k,m} \end{aligned} \tag{2}$$

The coefficient $a_{k,m} = \binom{k}{m}\binom{M - k}{M - m} \Big/ \binom{2M}{M}$ models the probability of transitioning from a state with $k$ mutants and $M - k$ wildtype to a state with $m$ mutants and $M - m$ wild type via the segregation of $2M$ mitochondria into two daughter cells with $M$ mitochondria each; the remaining part of the equation models the probability of reaching a state with $k$ mutant mitochondria through replication and mutation of $M$ mitochondria, of which $n$ are mutant (this corresponds to the probability of introducing $k - n$ new mutations). The system is updated $\overrightarrow{p}^{(2)} = Q^q \times \overrightarrow{p}^{(1)}$, across $q$ rounds of PGC cell division. We then apply particular processes to capture the effects of the bottleneck, follicular atresia, and cytoplasmic transfer.

As before, we model the bottleneck as $b$ extra rounds of segregation before the onset of mtDNA replication, following *Equation (1)* with $q + b$ cell divisions. This has no effect on the mean mutational number but increases mutational variance between the resulting PGCs. The transition between oogonia and primary oocytes occurs at random, and so does not alter the frequency distribution of mutants. Finally, during oocyte maturation, the mtDNA content of each cell doubles at every time step until the initial ploidy $M_0$ is restored. The transition matrix $G_{mn}$ is analogous to the first term of *Equation (2)*, incorporating replication and mutation, but without segregation (last term of *Equation (2)*):

$$G_{mn}^{(t)} = \binom{M^{(t)} - m}{n - m} U^{n-m}(1 - U)^{M-k} \tag{3}$$

*Equation (3)* models the probability of transitioning from a state with $m$ to a state with $n$ mutants, which is equivalent to the probability that exactly $n - m$ out of $M^{(t)} - m$ wildtype acquire a deleterious mutation. As the bottleneck reduces mtDNA copy number per cell, there is the need for $b$ extra

rounds of replication of mtDNA during oocyte maturation. Hence, the transition coefficient $G$ is applied $b + 12$ times in the bottleneck model, to restore the number of mtDNA copies per oocytes to the original ploidy level $M_0$: $\vec{p}^{(3)} = \left( \prod_t G^{(t)} \right) \times \vec{p}^{(2)}$.

At the end of the maturation phase, for the bottleneck model, selection is applied on individual fitness using a vector $w$ whose elements $w_m$ are equal to the corresponding fitness: $w_m = f(m) = 1 - \left( \frac{m}{M} \right)^5$. This causes a change in the population mutation load as the system is updated to:

$$\vec{p}^{(3)} = \left( \vec{Iw} \right) \vec{p}^{(2)} / \vec{w}^T \vec{p}^{(2)} \tag{4}$$

where $I$ is the identity matrix.

In the model of follicular atresia, an extra step is included to reflect selection that operates when the population of oogonia are culled to produce the primary oocytes. This causes a change in the population mutation load analogous to that described in *Equation (4)*, but using the cell fitness function $w_m = f(m) = 1 - \left( \frac{m}{M} \right)^\xi$ instead. This determines the shift in mutation loads that arises from fitness-dependent culling of oogonia. The transition coefficient *Equation (3)* for oocyte maturation is then applied 12 times in the follicular atresia model, to restore the original level of ploidy.

Finally, in order to model cytoplasmic transfer, a different process is used in the production of primary oocytes. A set of eight clonally derived cells is selected. The mutation levels of each cell in the cyst is obtained by applying *Equation (3)* three times. Then, 50% of the mitochondria in each cell are pooled into the Balbiani body of the primary oocyte. The probability for a cell with $m$ mutants to contribute $n$ mutants to the Balbiani body is given by:

$$C_{mn} = \binom{m}{n} p_{mut}^n \binom{M - m}{M/2 - n} p_{wt}^{M/2 - n} / N \tag{5}$$

which gives the number of permutations of $n$ mutant and $M/2 - n$ wild-type mtDNA copies, weighted by the probability of transfer $p_{mut}$ and $p_{wt}$ respectively, and divided by a normalization constant $N$. As the primary oocyte contains half of mitochondria from eight cells, it needs to undergo two fewer rounds of replication during oocyte maturation. Hence only 10 rounds of replication following *Equation (3)* are carried out in this case to restore the original level of ploidy.

For all three models (bottleneck, follicular atresia, and cytoplasmic transfer), the frequency distribution of mutation loads after these steps is used as the starting point for the next generation.

## Evolutionary dynamics and model accuracy

The processes described above are iterated until the Kullback-Leibler divergence (a theoretical measure of how two probability distributions differ from each other [*Kullback and Leibler, 1951*]) between the new and the old distribution is smaller than a threshold $\eta = 10^{-9}$. We then assume that the system has reached a stationary state, for example without significant changes in the overall distribution of mutation loads between generations (mutation-selection balance).

In order to compare the prediction of the model with the clinical data, we use the equilibrium distribution to calculate the fraction of the population which carries a detectable load of mitochondrial mutations but does not manifest any detrimental phenotype ($\alpha_1$) and the fraction of individuals affected by mitochondrial disease ($\alpha_2$) using a threshold of 60% mutation load to discriminate between carrier and disease status. Individuals are assumed to be mutation free beyond the detection threshold of 2% (*Elliott et al., 2008*).

The accuracy of the model is evaluated as the logarithm of the probability of reproducing clinical data by sampling the theoretical distribution at random. This is calculated as follows: let $X_1$ be the number of healthy individuals with detectable mutation load, and $X_2$ be the number of individuals affected by mitochondrial diseases; $N_1$ and $N_2$ the total number of individuals in the two trials; $\alpha_1$ and $\alpha_2$ the probability of observing, respectively, a healthy individual with detectable mutation load and an individual affected by mitochondrial disease, according to the prediction of the model. The log-likelihood of observing $X_1$ and $X_2$ by random sampling the theoretical distribution is given by

$$
\begin{aligned}
\log(\lambda_{tot}(\mu, M, \ldots)) &= \log\left[\prod_{i=1}^{2} p(X_i | \alpha_i(\mu, M, \ldots))\right] \\
&= \sum_{i=1}^{2} \log\left[\binom{N_i}{X_i} \alpha_i^{N_i} (1-\alpha_i)^{N_i - X_i}\right] \\
&= \sum_{i=1}^{2} \log\binom{N_i}{X_i} + N_i \alpha_i + (N_i - X_i)(1 - \alpha_i)
\end{aligned}
\tag{6}
$$

## Estimation of the deleterious mutation rate

The parameter values for the deleterious mutation rate we investigate reflect data collected from a number of species. Estimates of mtDNA point mutation rates in the crustacean *Daphnia pulex* range between $1.37 \times 10^{-7}$ and $2.28 \times 10^{-7}$ per site per generation (*Xu et al., 2012*). Assuming this rate applies to humans and there are ~20 cell divisions before oocyte maturation, leads to a range between $0.68 \times 10^{-8}$ and $1.14 \times 10^{-8}$ per site, per cell division. Analysis of *Caenorhabditis elegans* mtDNA leads to a similar estimate of ~$1.6 \times 10^{-7}$ per site, per generation (*Denver et al., 2000*), which corresponds to a rate of $0.8 \times 10^{-8}$ per site, per cell division. For *Drosophila melanoganster*, the mtDNA mutation rate yields an estimate of $6.2 \times 10^{-8}$ per site, per generation, and hence ~$0.31 \times 10^{-8}$ per site, per cell division (*Haag-Liautard et al., 2008*). Finally, analysis of human mtDNA point mutation rates give a mutation rate of 0.0043 per genome per generation (*Sigurğardóttir et al., 2000*), corresponding to ~$1.3 \times 10^{-8}$ mutations per site, per cell division.

These values do not take into account the presence of a number of processes likely to remove mutants and is therefore a conservative estimate. The loss of mutations would mean that the actual mutation rate is higher than the estimates above. But unlike nuclear rates, the compact structure of mtDNA where intergenic sequences are absent or limited to a few bases, means that the rate of point mutations is probably not much higher than the rate of deleterious mutations. Therefore, for this study, we consider a broad interval of possible deleterious mutation rates, labeled as low ($10^{-9}$), standard ($10^{-8}$) and high ($10^{-7}$).

## Acknowledgements

This work was supported by funding from the Engineering and Physical Sciences Research Council (EP/F500351/1, EP/I017909/1) and Natural Environment Research Council (NE/R010579/1) to AP, the Biotechnology and Biological Sciences Research Council (BB/S003681/1) and bgc3 to NL, and a joint grant to AP and NL from the Biotechnology and Biological Sciences Research Council (BB/V003542/1). We thank Molly Przeworski, David Rand and anonymous reviewer for their perceptive comments on the paper.

## Additional information

### Funding

| Funder | Grant reference number | Author |
| --- | --- | --- |
| Engineering and Physical Sciences Research Council | EP/F500351/1 | Andrew Pomiankowski |
| Engineering and Physical Sciences Research Council | EP/I017909/1 | Andrew Pomiankowski |
| Natural Environment Research Council | NE/R010579/1 | Andrew Pomiankowski |
| Biotechnology and Biological Sciences Research Council | BB/S003681/1 | Nick Lane |
| bgc3 | | Nick Lane |
| Biotechnology and Biological Sciences Research Council | BB/V003542/1 | Marco Colnaghi Andrew Pomiankowski Nick Lane |

The funders had no role in study design, data collection and interpretation, or the decision to submit the work for publication.

### Author contributions
Marco Colnaghi, Conceptualization, Software, Formal analysis, Investigation, Methodology, Writing - original draft, Writing - review and editing; Andrew Pomiankowski, Nick Lane, Conceptualization, Formal analysis, Supervision, Funding acquisition, Investigation, Methodology, Writing - original draft, Writing - review and editing

### Author ORCIDs
Marco Colnaghi (iD) https://orcid.org/0000-0002-5641-9324
Andrew Pomiankowski (iD) https://orcid.org/0000-0002-5171-8755
Nick Lane (iD) https://orcid.org/0000-0002-5433-3973

### Decision letter and Author response
Decision letter https://doi.org/10.7554/eLife.69344.sa1
Author response https://doi.org/10.7554/eLife.69344.sa2

## Additional files

### Supplementary files
• Transparent reporting form

### Data availability
All code has been posted on Github https://github.com/MarcoColnaghi1990/colnaghi-pomiankow-ski-lane-elife-2021 (copy archived at https://archive.softwareheritage.org/swh:1:rev:bdff29e963d2fe6cc915887539ffa4c04bea56b2).

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
