## [Decision Letter]

**Acceptance summary:**

Non-nuclear genomes, such as those of mitochondria, contribute to many aspects of cellular function, organismal function, and fitness. Understanding their biology and evolutionary dynamics is thus an essential component eukaryotic evolution. The manuscript addresses an important and complex problem regarding the relationship between mitochondrial mutations, their impacts on gamete function, and the attendant evolutionary processes. The authors present a computational approach to distinguish between three hypotheses about the level of selection most likely to explain the distribution of mitochondrial mutations in human populations. They propose that selection among mitochondria is the most likely process to match empirical, clinical data, for mitochondrial mutation loads. In the revised version the authors have taken care to bring the focus to mammalian systems, which now sits more comfortably with the model.

**Decision letter after peer review:**

Thank you for submitting your article "The need for high-quality oocyte mitochondria at extreme ploidy dictates germline development" for consideration by *eLife*. Your article has been reviewed by 2 peer reviewers, and the evaluation has been overseen by a Reviewing Editor and Molly Przeworski as the Senior Editor. The following individual involved in review of your submission has agreed to reveal their identity: David Rand (Reviewer #2).

Essential revisions:

All of us agree that the work is suitable for *eLife*, but there is one significant issue that requires attention. The issue arises from the fact that many of the biological assumptions and parameter choices for construction of your model are based on data from different organisms whose germ line biology is not comparable, and on a number of arguments that are drawn from older literature, but that are, in the light of current data, oversimplified to the point of creating "straw man"-like hypotheses. In short, there needs to be a better alignment between the model and the data. This is especially so when it comes to Figure 5 where you connect firmly and exclusively to data from humans.

I see two ways forward: 1, confine your model to human mitochondria and change the title of the paper; 2, retain the current title, re-think the content of the paper and possibly too, the model.

You will find that both reviewers have produced extensive and detailed critiques that will be invaluable in preparing a revised version. You should pay attention to all points raise.

*Reviewer #1 (Recommendations for the authors):*

This is a well written study with clear motivation to address an important and complex problem in multicellular eukaryotic biology. I suggest that this work would be strengthened by either (a) focusing the model, its parameter choices and interpretations of its findings, to human biology only, or (b) making it much more explicit which parameter values are drawn from which organisms, and explaining fully the many relevant well documented differences between PGG biology, mitochondrial behaviour, and Balbiani formation mechanisms, that exist across the wide range of organisms that the cited literature covers. A second suggestion to strengthen the manuscript is to replace some outdated or oversimplified strong generalizations (e.g. "plants do not have a germline" or "mitochondria are inherited exclusively maternally") with a fuller explanation of the complexity of existing observations, and a broader and more nuanced discussion of the multiple possible interpretations of the outcomes of their model. Specific suggestions to address these points are as follows:

1. The title is too strong given the observations presented. The authors present fit of a number of models to some epidemiological data, not empirical evidence that there are functional consequences to the "germline" (too vague) of mitochondrial "performance" in oocytes.

2. The authors focus the interpretation of their model on the Balbiani body, but their model appears to be suitable to other mechanisms of mitochondrial concentration in oocytes/gametogonia, as occurs in some animals at stages following Balbiani body dissolution. In other words, the model of selective accumulation of mitochondria from neighbouring/interconnected germ cells into oocyte precursors need not be restricted to a Balbiani body-mediated mechanism.

3. The authors should either restrict their model and interpretations to human biology, or be much more explicit about which organisms' biology is being incorporated into the various assumptions and interpretations throughout the manuscript. In some places the language and observations imply human biology or are clearly based on human biology, but this is not always made explicit and sometimes these things are extended or generalized to other organisms. Specific examples include the following:

a. Line 91 – "foetus" clearly implies human but the references cited here for "many animal species" include organisms where the "massive" germ cell number decline is not in the embryonic stage but rather in postembryonic stages, and organisms where the specific roles of the PGC descendants are not always the same (e.g. in *Drosophila* the germ cell descendants called nurse cells never had the potential to become oocytes, unlike many of the follicles that are destroyed in human ovaries).

b. Line 98 – "nurse cells," if the authors mean here PGC descendants that ultimately do not become gametes, do not serve the same biological functions across animals. The specific role of selective mitochondrial donation that the authors are concerned with here, may or may not be common in other such "nurse cell" systems, but the indispensable axial patterning roles that they play in other systems. Related to this point, The specific type of follicular atresia that humans display is not "almost universally conserved in the female germline of animal taxa" (lines 127-128).

c. Line 96 – "menopause" does not apply to all animals, including many of the animals whose biology is used throughout to inform the model and its interpretations (e.g. *Drosophila*, zebrafish, *Xenopus*, Thermobia).

d. Line 109 – "clusters of 5-8 cells" – germ cell cysts (interconnected groups of cells that are the clonal descendants of a single germ cell, some or all of which may go on to complete gamete differentiation) can consist of fewer than five or of more than eight cells depending on the animal, and in some species (including *Mus musculus*, which the authors frequently use as a source of information) the number does not appear fixed even within an individual (see Lei and Spradling, 2016, their citation #41).

e. The blanket statement that "plants and basal metazoa" do not sequester a germ line nor have oocytes with mitochondrial enrichment is not supported by the empirical evidence and not even by citation 67 provided here, which explicitly discusses the clear empirical evidence that flowering plants *do* have a specific gametogenic cell population that is distinct from differentiated somatic cells. Neither of the two citations here provides any data on sponges, corals or placozoans (correct spelling), which in any case are not "basal." There are many examples in the empirical literature that show that at least some Porifera and Cnidaria do have unique gametogenic populations that are segregated in embryogenesis, and that not any somatic cell is capable of gametogenesis.

f. Clarify in Figure 1 and Figure 4 which animals' biology this schematic is based on

4. There are a number of instances where the biological phenomena described by the authors or modeled in their simulations is either an overgeneralization or an overinterpretation of existing data, even data within the citations referenced by the authors, or is not supported by the provided citations. These should be re-written to more precisely and neutrally describe previous observations and their relevance to the model parameters and interpretations of this study, and citations should be corrected to reflect the actual primary data source of the claim. Specific examples include the following:

a. Line 110 "mitochondria are…streamed into the Balbiani body" – the citations listed in support of this statement (41, 47) do not show active movement of mitochondria into Balbiani bodies. Rather, they show evidence for different numbers of mitochondria in oocytes and non-oocyte-destined germ cells in mouse ovaries at different stages. The citations offered in support of "cytoplasmic transfer" in the following sentence (48, 49) likewise do not provide data demonstrating this, but rather are reviews discussing the hypothesis that the authors favor in their study. Examples of active transport of mitochondria along polymerized cytoskeletal proteins cited later on in the manuscript (e.g. 50) do not demonstrate that this takes place in oocytes bur rather study cultured human (HeLa) cells.

b. Line 161 – citation 57 is another modeling paper by some of the authors of the current study; it is not a primary data source supporting the number of mitochondria or mitochondrial genomes in mature (human? unclear) oocytes as implied.

c. Line 269 – citation 61 provided for the Balbiani body being "a nearly universal feature of female germlines in animals" is a book about insect ovaries only.

d. Lines 270-271 – cysts of 8 cells are not "typical in mammalian development" and the citations provided here (41, 47) not only deal only with mice (and not other mammals), but furthermore demonstrate that even within the ovaries of the same species, cysts do not all have the same number of cells in them.

5. Related to the point above, the model has connections between cells in a cyst forming when there are 8 cells – in other words, their model has cells divide with complete cytokineses, followed by *de novo* bridge formation between cells. However, in most documented animals that I am aware of that form germ cell cysts during gameotgenesis, the mitoses of cyst-forming divisions are incomplete, omitting cytokineses such that the cytoplasm of cells of the cyst is continuously connected even before the final number of cells of the cyst is achieved. Thus, cytoplasmic organelles can be and sometimes are (e;g. the fusome in the case of *Drosophila*) transferred between cells of the cyst continuously, rather than at a single defined stage as defined in this model. Furthermore, a consequence of the way that cysts form is that not all cells of the cysts have the same number of connections to other cells in the cyst, and the possibility of unequal movement between or access to cells of the cyst should therefore be considered (even if the authors decide to eliminate this potential from their model, they should acknowledge this choice). Moreover, in mice, as documented by their citation 41, cytoplasmic bridges are unlikely to be the only way that mitochondria may move between cells, as the authors of that work document simply ruptures in the cell membrane of neighbouring cells, which may not even be members of the same (clonal) cyst. The authors should clarify this aspect of their model and that it may introduce a deviation from how organelles like mitochondria may actually behave, or at a minimum include only some of the biological features that contribute to determining which mitochondria end up in the oocyte.

a. Similarly, the two final sentences of the discussion must be revised to qualify (a) that it is definitively *not* the case that all animals generate oocytes with cytoplasm from multiple cells – see for example the >12 orders of insects *without* nurse cells that are discussed in their citation 61; and (b) that it is also not the case that all animals derive their mitochondria exclusively maternally – there are well documented cases of paternal inheritance of mitochondria from multiple animals, and it has not, to my knowledge, been conclusively demonstrated that the mitochondria from sperm that can and do enter the egg with mammalian sperm make no contribution to the zygotic mitochondrial populations (but if the authors are aware of such definitive demonstration then they should provide it here).

6. Lines 311, 319 – here and elsewhere the authors need to make it very clear that the epidemiological data from "human populations" used to compare with the outcomes of the tested models is derived from only hundreds or a few thousands of adults in northeast England, published 13 years ago. If these data have subsequently been compared to those derived from larger and more diverse populations of humans from around the globe, the authors should make this clear.

7. Lines 330-331: the individual- and cell-level selection models are described as providing a "credible" (not defined quantitatively) explanation for demographic data "only at low mutation rates" and this section goes on to conclude that the Balbiani body-selection (BB) model is better at explaining the observed data. However, the same mutation rate (10E-8) that is here called "low" is then called "realistic" when considering the BB model (line 336) and "standard" in the discussion (line 455).

8. The normal mitotic program of germ cells in some animals is frequently referred to as "over-proliferation." The authors should eliminate this term, which implies something "unnecessary," with something more neutral that simply describes the biology. The degree of cell proliferation and cell death are usually, and definitely in the case of the germ line, tightly regulated developmental processes. The fact that not all progeny of the PGCs end up creating gametes is common in many animals and does not mean that PGCs that do not create gametes serve no biological function. The repeated description of this proliferation program as a "longstanding paradox" or "long….puzzling" in this manuscript therefore does not reflect a modern understanding of the developmental process, in which temporary tissues are often created and subsequently destroyed in both the germ line and in the soma. In this context the unfounded speculation that "such a high proportion of germ cells could have low fitness" (line 413) does not make sense, and much of the final paragraph of the discussion (line 477 – 502), which is entirely without backing evidence, should be at a minimum significantly rewritten with citations provided demonstrating evidence for the many claims therein, if not (preferably) removed from the manuscript entirely.

*Reviewer #2 (Recommendations for the authors):*

The paper presents an advance of interest to the general readers of *eLife*.

Line 182: What effect does mitochondrial growth during this PGC phase have on the model outcomes? Is it true that no growth happens? It seems that it could in other organisms, so adding that possibility will increase the generality of the model.

Line 190: As a follow up to the possibility of replication during growth, will mutation during PGC proliferation generate a different variance in mutations than mitochondrial proliferation in non-dividing mature oocytes? The differences in these distributions could affect the efficacy of selection against mtDNA, mitos, cells, and individuals. It seems like the old school fluctuation analyses of estimating mutations in growing bacterial populations might have an analogy here.

Line 260 and Figure 3. Figure 3C has epistasis listed as 1, 2, 3, not 1, 2, 5 ?

Lines 298-301. The yellow shaded areas in 4B look a lot like those in Figure 1B in terms of size and shape, but both are distinctly different from those in the other figures. Is there a reason for this given different parameters used in these versions of the model (Figure 1 vs 4)?

Lines 338-339 and Figure 5. One additional component of the model that should be considered is to estimate the effect of selection on the somatic cell divisions in tissues that are used to estimate mutations loads in humans. If functional mitochondria are selected for in post zygotic or post embryonic development, a liver biopsy, or a blood sample, or a post-mortem brain sample may have filtered the mtDNA population such that the modeling of the germ cells is not faithfully represented in adult somatic tissues. Some simple additions of the iterations of the model, post zygotically, could elucidate whether this is an important effect, or not. It is relevant, since heteroplasmy levels differ among tissues in humans, so the matching with clinical data could allow for a pick-and-choose target in the clinical data. This is yet another controversy whether tissue specific heteroplasmy levels is due to early developmental effects in tissue determination, or the effects of among-cell or among mito selection during tissue turnover through adulthood.

Lines 397-403. These post zygotic effects could change the relative importance of individual-mitochondria vs. individual-cell selection, since somatic proliferation and cellular turnover will not involve the important cytoplasmic bridge effects of the Balbiani cloud in typical somatic tissue differentiation (and hence the tissue used to estimate mutation load in clinical surveys).

Line 524. 'Imput' should be input.

Lines 537-541. How does the epistasis factor play into the different degree of purifying selection across nucleotides in mtDNA? Synonymous sites have 10-30x faster substitution rates, but amino acid sites have slower substitution rates, than many nuclear proteins. Substitution is not equal to mutation, but will the fitness effect of mutations accumulate statistically in different ways if all sites, vs. only coding sites, are subject to epistasis. The target size (g) of mutational input is the issue here.

---

## [Author Response]

Essential revisions:All of us agree that the work is suitable for eLife, but there is one significant issue that requires attention. The issue arises from the fact that many of the biological assumptions and parameter choices for construction of your model are based on data from different organisms whose germ line biology is not comparable, and on a number of arguments that are drawn from older literature, but that are, in the light of current data, oversimplified to the point of creating "straw man"-like hypotheses. In short, there needs to be a better alignment between the model and the data. This is especially so when it comes to Figure 5 where you connect firmly and exclusively to data from humans.I see two ways forward:1) Confine your model to human mitochondria and change the title of the paper.2) Retain the current title, re-think the content of the paper and possibly too, the model.

Thank you for this advice. We have changed the title to "….. mammalian germline development". This identifies the main area for which we have data.

We comment elsewhere on other non-mammalian systems to point to generalities. This is your #1 but extended to humans and mice. Other areas in the Introduction where we circumscribe our approach are:

Abstract limits attention to mouse and humans;

Line 30 – “Mutation rates and germline development parameters from mouse and humans”;

Line 92 – Rather than saying follicular atresia occurs in many animal species including humans, we now explain follicular atresia in humans, and then comment that gamete loss is a feature also of mice and many other animal germlines;

Line 108 – When we introduce the Balbiani body we say it is a prominent feature of human and mouse female germline, as well as a range of other invertebrates and vertebrates.

You will find that both reviewers have produced extensive and detailed critiques that will be invaluable in preparing a revised version. You should pay attention to all points raise.Reviewer #1 (Recommendations for the authors):This is a well written study with clear motivation to address an important and complex problem in multicellular eukaryotic biology. I suggest that this work would be strengthened by either (a) focusing the model, its parameter choices and interpretations of its findings, to human biology only, or (b) making it much more explicit which parameter values are drawn from which organisms, and explaining fully the many relevant well documented differences between PGG biology, mitochondrial behaviour, and Balbiani formation mechanisms, that exist across the wide range of organisms that the cited literature covers. A second suggestion to strengthen the manuscript is to replace some outdated or oversimplified strong generalizations (e.g. "plants do not have a germline" or "mitochondria are inherited exclusively maternally") with a fuller explanation of the complexity of existing observations, and a broader and more nuanced discussion of the multiple possible interpretations of the outcomes of their model. Specific suggestions to address these points are as follows:1. The title is too strong given the observations presented. The authors present fit of a number of models to some epidemiological data, not empirical evidence that there are functional consequences to the "germline" (too vague) of mitochondrial "performance" in oocytes.

We have changed the title to focus on mammalian systems, humans and mice.

2. The authors focus the interpretation of their model on the Balbiani body, but their model appears to be suitable to other mechanisms of mitochondrial concentration in oocytes/gametogonia, as occurs in some animals at stages following Balbiani body dissolution. In other words, the model of selective accumulation of mitochondria from neighbouring/interconnected germ cells into oocyte precursors need not be restricted to a Balbiani body-mediated mechanism.

It is not clear to us what the referee is alluding to. Literature is often bedevilled by terminology invented in specific study systems. This can't be addressed in this manuscript, but would be appropriate for a review, drawing together disparate literature.

The Balbiani body has been identified in mice, more recently humans, and more widely in frogs, clams, *Drosophila* and Thermobia, a primitive insect. In all these systems it is thought to be a pooling mechanism for cytoplasm. We allude to this generality (line 108-) but concentrate the work on human/mouse. We are not sure what other mechanism the referee has in mind.

We deliberately phrase our modelling in a general fashion based on the level at which selection takes place: individual, cell, organelle. This should lead readers, including experts like the reviewer, to see where the ideas might apply in other study systems. We tie it to particular instances, but agree that it would generalise.

3. The authors should either restrict their model and interpretations to human biology, or be much more explicit about which organisms' biology is being incorporated into the various assumptions and interpretations throughout the manuscript. In some places the language and observations imply human biology or are clearly based on human biology, but this is not always made explicit and sometimes these things are extended or generalized to other organisms. Specific examples include the following:a. Line 91 – "foetus" clearly implies human but the references cited here for "many animal species" include organisms where the "massive" germ cell number decline is not in the embryonic stage but rather in postembryonic stages, and organisms where the specific roles of the PGC descendants are not always the same (e.g. in *Drosophila* the germ cell descendants called nurse cells never had the potential to become oocytes, unlike many of the follicles that are destroyed in human ovaries).

Thanks. We have changed the wording, describing humans first, then generalising to other species "Similar loss of female germ cells before sexual maturity is evident in mice and several other animal species " (line 92) – which leads away from specifying exactly when in development the germ cell loss occurs in individual species.

b. Line 98 – "nurse cells," if the authors mean here PGC descendants that ultimately do not become gametes, do not serve the same biological functions across animals. The specific role of selective mitochondrial donation that the authors are concerned with here, may or may not be common in other such "nurse cell" systems, but the indispensable axial patterning roles that they play in other systems. Related to this point, The specific type of follicular atresia that humans display is not "almost universally conserved in the female germline of animal taxa" (lines 127-128).

The referee is reading more into our statement. We claim that germ-cell loss is universal; we do not say that the human system is also seen in other species. But to avoid unnecessary contention we remove the words "almost universal" and modify the wording to "…germ-cell over-proliferation followed by massive loss which is a widely conserved feature of the female germline in animal taxa" (line 129-131)

c. Line 96 – "menopause" does not apply to all animals, including many of the animals whose biology is used throughout to inform the model and its interpretations (e.g. *Drosophila*, zebrafish, *Xenopus*, Thermobia).

Agreed – now only attached to humans.

d. Line 109 – "clusters of 5-8 cells" – germ cell cysts (interconnected groups of cells that are the clonal descendants of a single germ cell, some or all of which may go on to complete gamete differentiation) can consist of fewer than five or of more than eight cells depending on the animal, and in some species (including *Mus musculus*, which the authors frequently use as a source of information) the number does not appear fixed even within an individual (see Lei and Spradling, 2016, their citation #41).

Agreed. We limit the statement to mice, and to be more precise say " typically form clusters of 5-8 cells " (line 110-111).

e. The blanket statement that "plants and basal metazoa" do not sequester a germ line nor have oocytes with mitochondrial enrichment is not supported by the empirical evidence and not even by citation 67 provided here, which explicitly discusses the clear empirical evidence that flowering plants do have a specific gametogenic cell population that is distinct from differentiated somatic cells. Neither of the two citations here provides any data on sponges, corals or placozoans (correct spelling), which in any case are not "basal." There are many examples in the empirical literature that show that at least some Porifera and Cnidaria do have unique gametogenic populations that are segregated in embryogenesis, and that not any somatic cell is capable of gametogenesis.

Not wanting to get caught up in controversy about early diverging metazoans, we have substituted simple for basal. But this may be objectionable as well. We now say: This contrasts with organisms that have modular growth, such as plants and morphologically simple metazoa (sponges, corals, placozoa), which neither sequester a recognizable germline distinct from the stem-cell lineage early in development (although recent work challenges this view), nor have oocytes with massively expanded mitochondrial numbers. We have amended the references. Line 416-9.

f. Clarify in Figure 1 and Figure 4 which animals' biology this schematic is based on.

Inserted the qualifier "human" in Figure 1 "Timeline of human oocyte development showing the main stages modelled". When first mentioned in the text (line 136), humans are identified already as the relevant model species for Figure 1. This carries over to Figures 2-4, clearly extrapolations. And Figure 5 already mentions humans.

4. There are a number of instances where the biological phenomena described by the authors or modeled in their simulations is either an overgeneralization or an overinterpretation of existing data, even data within the citations referenced by the authors, or is not supported by the provided citations. These should be re-written to more precisely and neutrally describe previous observations and their relevance to the model parameters and interpretations of this study, and citations should be corrected to reflect the actual primary data source of the claim. Specific examples include the following:a. Line 110 "mitochondria are…streamed into the Balbiani body" – the citations listed in support of this statement (41, 47) do not show active movement of mitochondria into Balbiani bodies. Rather, they show evidence for different numbers of mitochondria in oocytes and non-oocyte-destined germ cells in mouse ovaries at different stages. The citations offered in support of "cytoplasmic transfer" in the following sentence (48, 49) likewise do not provide data demonstrating this, but rather are reviews discussing the hypothesis that the authors favor in their study. Examples of active transport of mitochondria along polymerized cytoskeletal proteins cited later on in the manuscript (e.g. 50) do not demonstrate that this takes place in oocytes bur rather study cultured human (HeLa) cells.

Thank you for your attention to this detail. We preface our remarks with "It is thought that…." (line 110) and similar (line 441). Our reference to [50] carefully refers to mitosis (line 137).

b. Line 161 – citation 57 is another modeling paper by some of the authors of the current study; it is not a primary data source supporting the number of mitochondria or mitochondrial genomes in mature (human? unclear) oocytes as implied.

This is appropriate as it is in the description of the modelling section – so it refers to another modelling paper, making a similar assumption. This is a model assumption, and is clearly flagged as "Computational Model".

c. Line 269 – citation 61 provided for the Balbiani body being "a nearly universal feature of female germlines in animals" is a book about insect ovaries only.

Oops – that is wrong, and unnecessary. So removed.

d. Lines 270-271 – cysts of 8 cells are not "typical in mammalian development" and the citations provided here (41, 47) not only deal only with mice (and not other mammals), but furthermore demonstrate that even within the ovaries of the same species, cysts do not all have the same number of cells in them.

Thanks, again this is wrong, unnecessary and has been removed.

5. Related to the point above, the model has connections between cells in a cyst forming when there are 8 cells – in other words, their model has cells divide with complete cytokineses, followed by de novo bridge formation between cells. However, in most documented animals that I am aware of that form germ cell cysts during gameotgenesis, the mitoses of cyst-forming divisions are incomplete, omitting cytokineses such that the cytoplasm of cells of the cyst is continuously connected even before the final number of cells of the cyst is achieved. Thus, cytoplasmic organelles can be and sometimes are (e;g. the fusome in the case of *Drosophila*) transferred between cells of the cyst continuously, rather than at a single defined stage as defined in this model. Furthermore, a consequence of the way that cysts form is that not all cells of the cysts have the same number of connections to other cells in the cyst, and the possibility of unequal movement between or access to cells of the cyst should therefore be considered (even if the authors decide to eliminate this potential from their model, they should acknowledge this choice). Moreover, in mice, as documented by their citation 41, cytoplasmic bridges are unlikely to be the only way that mitochondria may move between cells, as the authors of that work document simply ruptures in the cell membrane of neighbouring cells, which may not even be members of the same (clonal) cyst. The authors should clarify this aspect of their model and that it may introduce a deviation from how organelles like mitochondria may actually behave, or at a minimum include only some of the biological features that contribute to determining which mitochondria end up in the oocyte.

A model cannot possibly capture all reality. The central principle of the model is not undermined by these details. Namely – mitochondria are pooled and selection operates at the organelle level. In our opinion, we cannot review all these intricacies, nor would they add to our general message. When we first discuss the Balbiani body in the Introduction, we do not specify closely when the transfer is achieved.

a. Similarly, the two final sentences of the discussion must be revised to qualify (a) that it is definitively not the case that all animals generate oocytes with cytoplasm from multiple cells – see for example the >12 orders of insects without nurse cells that are discussed in their citation 61; and (b) that it is also not the case that all animals derive their mitochondria exclusively maternally – there are well documented cases of paternal inheritance of mitochondria from multiple animals, and it has not, to my knowledge, been conclusively demonstrated that the mitochondria from sperm that can and do enter the egg with mammalian sperm make no contribution to the zygotic mitochondrial populations (but if the authors are aware of such definitive demonstration then they should provide it here).

This is strange nit-picking. Our final two sentences address the claim made by Buss, and repeated elsewhere, that to avoid selfish conflict, there must be derivation of gametes from a single cell. This is not the case for mitochondria and that is a very interesting general observation. The referee reads something different – a claim that all animals have Balbiani bodies. It would be perverse for the last sentence to include qualifiers to something that we are not saying. We could insert "generally" in each sentence – but what would that serve (but have done so for uniparental inheritance)

6. Lines 311, 319 – here and elsewhere the authors need to make it very clear that the epidemiological data from "human populations" used to compare with the outcomes of the tested models is derived from only hundreds or a few thousands of adults in northeast England, published 13 years ago. If these data have subsequently been compared to those derived from larger and more diverse populations of humans from around the globe, the authors should make this clear.

This is all there is, as far as we know.

7. Lines 330-331: the individual- and cell-level selection models are described as providing a "credible" (not defined quantitatively) explanation for demographic data "only at low mutation rates" and this section goes on to conclude that the Balbiani body-selection (BB) model is better at explaining the observed data. However, the same mutation rate (10E-8) that is here called "low" is then called "realistic" when considering the BB model (line 336) and "standard" in the discussion (line 455).

We agree with the referee and have tightened up our language:

1. The word credible has particular meaning in statistics, so has been removed, and the sentence re-written.

2. We use low (10-9), standard (10-8) and high (10-7) as terms for particular mutation rates, which are set out in the computational model section (line 154) and in the Methods (line 767-8).

8. The normal mitotic program of germ cells in some animals is frequently referred to as "over-proliferation." The authors should eliminate this term, which implies something "unnecessary," with something more neutral that simply describes the biology. The degree of cell proliferation and cell death are usually, and definitely in the case of the germ line, tightly regulated developmental processes. The fact that not all progeny of the PGCs end up creating gametes is common in many animals and does not mean that PGCs that do not create gametes serve no biological function. The repeated description of this proliferation program as a "longstanding paradox" or "long….puzzling" in this manuscript therefore does not reflect a modern understanding of the developmental process, in which temporary tissues are often created and subsequently destroyed in both the germ line and in the soma. In this context the unfounded speculation that "such a high proportion of germ cells could have low fitness" (line 413) does not make sense, and much of the final paragraph of the discussion (line 477 – 502), which is entirely without backing evidence, should be at a minimum significantly rewritten with citations provided demonstrating evidence for the many claims therein, if not (preferably) removed from the manuscript entirely.

We disagree with the referee. The phrase over-proliferation followed by massive loss captures what is going on. There is no statement that this is not regulated. We do not say that this serves no function. There is a puzzle to explain.

The referee comments here are a matter of their personal style. They are entitled to write their own work in the manner they prefer, but should not push us into their own style.

Reviewer #2 (Recommendations for the authors):The paper presents an advance of interest to the general readers of eLife.Line 182: What effect does mitochondrial growth during this PGC phase have on the model outcomes? Is it true that no growth happens? It seems that it could in other organisms, so adding that possibility will increase the generality of the model.

For simplicity we allow the number of mitochondria in PGCs to remain constant throughout the period of PGC proliferation. So the "bottleneck size" remains consistent for this period. We add a comment that this is an assumption (line 186). A transitory bottleneck where numbers went down and up again would have relatively less effect. We comment on this in the Discussion as it remains an area where there are different results in the literature (line 391-5).

The referee asks for generality across species which we wish we could give. But there is little distinct evidence from other species about the number of mtDNA copies during development.

Line 190: As a follow up to the possibility of replication during growth, will mutation during PGC proliferation generate a different variance in mutations than mitochondrial proliferation in non-dividing mature oocytes? The differences in these distributions could affect the efficacy of selection against mtDNA, mitos, cells, and individuals. It seems like the old school fluctuation analyses of estimating mutations in growing bacterial populations might have an analogy here.

It does. With relatively low numbers and cell division in PGCs, mutational variation can build up considerably as stochastic number change occurs in daughter cells. This does not happen to such a great extent in maturing oocytes where there is stochastic accumulation of mutation in mitochondria but no stochastic segregation at cell division.

Line 260 and Figure 3. Figure 3C has epistasis listed as 1, 2, 3, not 1, 2, 5 ?

Thanks, corrected.

Lines 298-301. The yellow shaded areas in 4B look a lot like those in Figure 1B in terms of size and shape, but both are distinctly different from those in the other figures. Is there a reason for this given different parameters used in these versions of the model (Figure 1 vs 4)?

We have improved the description in the legend to make clear where differences with the base model are evident.

Lines 338-339 and Figure 5. One additional component of the model that should be considered is to estimate the effect of selection on the somatic cell divisions in tissues that are used to estimate mutations loads in humans. If functional mitochondria are selected for in post zygotic or post embryonic development, a liver biopsy, or a blood sample, or a post-mortem brain sample may have filtered the mtDNA population such that the modeling of the germ cells is not faithfully represented in adult somatic tissues. Some simple additions of the iterations of the model, post zygotically, could elucidate whether this is an important effect, or not. It is relevant, since heteroplasmy levels differ among tissues in humans, so the matching with clinical data could allow for a pick-and-choose target in the clinical data. This is yet another controversy whether tissue specific heteroplasmy levels is due to early developmental effects in tissue determination, or the effects of among-cell or among mito selection during tissue turnover through adulthood.

This is a great suggestion for a future research project. We entirely agree that there remains a very interesting question to address here about development and levels of heteroplasmy across different tissues. We obliquely addressed this in a prior paper on the origin of the germline (Radzvilavicius et al., 2016 PLoS-B). There is a lot of work to do here, and it would be too much to try and include an analysis in this paper. But we have added a couple of sentences alluding to this in the Discussion (line 507-12)

Lines 397-403. These post zygotic effects could change the relative importance of individual-mitochondria vs. individual-cell selection, since somatic proliferation and cellular turnover will not involve the important cytoplasmic bridge effects of the Balbiani cloud in typical somatic tissue differentiation (and hence the tissue used to estimate mutation load in clinical surveys).

Good point. But see answer above.

Line 524. 'Imput' should be input.

Done.

Lines 537-541. How does the epistasis factor play into the different degree of purifying selection across nucleotides in mtDNA? Synonymous sites have 10-30x faster substitution rates, but amino acid sites have slower substitution rates, than many nuclear proteins. Substitution is not equal to mutation, but will the fitness effect of mutations accumulate statistically in different ways if all sites, vs. only coding sites, are subject to epistasis. The target size (g) of mutational input is the issue here.

We set this up in the Introduction. It is difficult to address briefly. If there is epistasis between mitochondrial mutations, it is likely to retard mutation accumulation at selected sites. There will also be expected changes in heteroplasmy – and that's not something we've thought about, and we don't think this has been addressed in the theoretical or empirical literature (and is not really the theme here).

It may also be that we should limit the estimate of g to the number of "non-neutral sites", rather than assume as we did that all sites are potentially subject to selection. We now acknowledge this as an assumption.